# Implications of surface flooding on airborne estimates of snow depth on sea ice

Anja Rösel[1,6*], Sinead Louise Farrell[2], Vishnu Nandan[3], Jaqueline Richter-Menge[4], Gunnar Spreen[5,1], Dmitry V. Divine[1], Adam Steer[1], Jean-Charles Gallet[1], and Sebastian Gerland[1]

[1]Norwegian Polar Institute, Fram Centre, Tromsø, Norway
[2]Department of Geographical Sciences, University of Maryland, College Park, MD, USA
[3]Centre for Earth Observation Science (CEOS), University of Manitoba, MB, Canada
[4]University of Alaska Fairbanks, Fairbanks, AK, USA
[5]Institute of Environmental Physics, University of Bremen, Bremen, Germany
[6] now at Remote Sensing Technology Institute, German Aerospace Center (DLR), Wessling, Germany

*Correspondence to*: Anja Rösel (anja.roesel@dlr.de)

**Abstract.** Snow depth observations from airborne snow radars, such as the NASA's Operation IceBridge (OIB) mission, have recently been used in altimeter-derived sea ice thickness estimates, as well as for model parameterization. A number of validation studies comparing airborne and in-situ snow depth measurements have been conducted in the western Arctic Ocean, demonstrating the utility of the airborne data. However, there have been no validation studies in the Atlantic sector of the Arctic. Recent observations in this region suggest a significant and predominant shift towards a *snow-ice* regime, caused by deep snow on thin sea ice. During the Norwegian young sea ICE expedition (N-ICE2015) in the area north of Svalbard, a validation study was conducted on March 19, 2015. This study collected ground truth data during an OIB overflight. Snow and ice thickness measurements were obtained across a two dimensional (2-D) 400 m × 60 m grid. Additional snow and ice thickness measurements collected in-situ from adjacent ice floes helped to place the measurements obtained at the gridded survey field site into a more regional context. Widespread negative freeboards and flooding of the snowpack were observed during the N-ICE2015 expedition, due to the general situation of thick snow on relatively thin sea ice. These conditions caused brine wicking into and saturation of the basal snow layers. This causes the airborne radar signal to undergo more diffuse scattering, resulting in the location of the radar main scattering horizon to be detected well above the snow/ice interface. This leads to a subsequent underestimation of snow depth, if only radar-based information is used, the average airborne snow depth was 0.16 m thinner than that measured in-situ at the 2-D survey field. Regional data within 10 km of the 2-D survey field suggested however a smaller deviation between average airborne and in-situ snow depth, a 0.06 m underestimate in snow depth by the airborne radar, which is close to the resolution limit of the OIB snow radar system. Our results also show a broad snow depth distribution, indicating a large spatial variability in snow across the region.

Differences between the airborne snow radar and in-situ measurements fell within the standard deviation of the in-situ data (0.15 – 0.18 m). Our results suggest that sea water flooding of the snow/ice interface leads to underestimations in snow depth or overestimations of sea ice freeboard, measured from radar altimetry, in turn impacting the accuracy of sea ice thickness estimates.

## 1 Introduction

Snow and sea ice thickness in a changing Arctic climate system are the matter of many recent studies (e.g. Webster et al. 2018), since the snow layer on top of the frozen ocean generates several contradictory effects on the polar climate. On the one hand, in winter, snow acts as an insulator between the relatively warm ocean and the cold atmosphere and hinders the heat exchange between ocean and atmosphere, reducing the sea ice growth rate (Sturm, 2002; Perovich, 2003). On the other hand, in spring and summer, snow reflects with its high optical albedo in the range of 0.7-0.85 short-wave radiation and prevents the underlying sea ice with an albedo of about 0.6 from melting (Grenfell and Maykut, 1977, Perovich, 1996). In addition, snow cover controls the amount of transmittance of photosynthetically active radiation affecting the productivity of primary algae and phytoplanktons (Mundy et al., 2007). Moreover, snow can be a positive contributor to the sea ice mass balance since snow can transform to snow-ice (Granskog et al., 2017; Merkouriadi et al., 2017a) and superimposed ice (Eicken et al., 2004; Wang et al., 2015).

Besides the importance of snow from a radiative and mass balance perspective, knowledge of snow depth on sea ice is also required for the accurate retrieval of sea ice thickness from satellite altimetry. The method relies on the assumption that sea ice floating in the ocean is in hydrostatic equilibrium, and sea ice thickness can be calculated by using observations of either ice-freeboard (from radar altimeters) or snow-freeboard (from laser altimeters) and assumptions about the respective densities of snow, ice and water. Ice- and snow-freeboard describe the distances above the local sea level to the snow/ice or air/snow interface, respectively. The error budget of the derived ice thickness from laser altimetry is dominated by uncertainties of snow depth and ice and snow densities, as well as uncertainties due to remaining errors in the sea surface height (Giles et al., 2007; Kern et al., 2015; Skourup et al., 2017).

Thus, accurate knowledge of snow depth on sea ice would be helpful to reduce the error in the sea ice thickness calculations and is important for quantifying climatological processes in Polar Regions. The Operation IceBridge (OIB) airborne campaigns (Koenig et al., 2010), which began in 2009, measure snow depth and surface elevation with an ultra-wideband snow radar (e.g., Yan et al., 2017) and an airborne topographic mapper (ATM) laser altimetry system (Krabill et al., 2002), respectively. With these sensors, both the air/snow and the snow/ice interface can be detected with the snow radar (e.g., Newman et al., 2014), and the surface elevation can be mapped with the ATM (e.g., Farrell et al., 2012). Hence, the OIB data are a valuable source for validating satellite remote sensing sea ice products as well as for model parameterization. Furthermore, the comparison of airborne OIB data with in-situ field measurements is necessary to understand the processes affecting radar penetration into snow covered sea ice and the impact of the snow load on the snow/ice interface.

Several OIB validation studies have been conducted (e.g., Farrell et al., 2012; Webster et al., 2014; Newman et al., 2014; Holt et al., 2015), multiple snow depth retrieval algorithms were developed (e.g., Kurtz et al., 2013, 2014; Newman et al., 2014; Kwok and Maksym, 2014), and compared with satellite products (Kwok et al., 2017; Lawrence et al., 2018). These studies have provided insights about the snow depth uncertainty and the errors associated with the airborne techniques (Kwok, 2014; King et al., 2015). However, in the northern hemisphere, all evaluation studies (except those connecting to satellite data) have thus far focused on snow in the Canada Basin, in the central Arctic Ocean, or only in peripheral sub regions of the Arctic. To our knowledge, no OIB validation study has been conducted in the Atlantic sector of the Arctic.

In recent years, a significant change towards thinner ice with thicker snow cover (Renner et al., 2014; Rösel et al., 2018) has been observed in this region, caused by an increase in intense storm events and associated precipitation in this area (Woods and Caballero, 2016; Graham et al., 2017; Rinke et al., 2017). In addition, previous studies indicate that radar signal penetration through the snow pack might be lower under certain geophysical snow-ice conditions in this area (Gerland et al., 2013; King et al., 2018; Nandan et al., 2020) and also in the Antarctic (Kwok and Kacimi, 2018; Willatt et al., 2010). Snow and ice conditions in this region differ to those in the Canada Basin and central Arctic (e.g., Webster et al., 2014; 2018), and they have been found to induce substantial negative ice freeboards with subsequent flooding of the snow pack, more akin to the conditions in the seasonal ice pack of the Southern Ocean (Massom et al., 2001). This may have an impact on remote sensing methods of snow and ice thickness estimation, which have so far only been validated for more typical Arctic conditions.

In this paper, we present in-situ observations of sea ice and snow depth, and snow and ice characteristics from the N-ICE2015 expedition, alongside near-coincident airborne measurements acquired on 19 March 2015 during an OIB overflight. We calculate ice freeboard values from a variety of sensors to investigate the prevalence of negative freeboards and flooding at the snow/ice interface. We investigate the impact of flooded snow layers on the airborne radar observations. Utilizing a combination of methodologies, we assess sea ice thickness conditions in the region. We discuss our results in the context of satellite-derived ice thickness and consider the impact of flooding on estimating thickness in regions with thin sea ice and deep snow, such as in the Atlantic section of the Arctic Ocean or in the Southern Ocean.

## 2 Data and Methods

### 2.1 Study Area

Field observations for this study were acquired during the Norwegian young sea ICE expedition (N-ICE2015) with the research vessel *RV Lance*. The expedition started in the Arctic Ocean north of Svalbard at 83°15'N, 21°32'E on 15 January 2015 and concluded at 80°N and 5°36'E on 22 June 2015 and consisted of a series of four drift segments (Granskog et al., 2016, 2018). In this study, we focus on sea ice and snow related observations from the drift of *Floe 2*, covering a time period from 24 February to 19 March 2015. Data from the OIB overflight employed in this study was collected on 19 March 2015

at 82°29'N and 22°37'E above the drifting sea ice floe.

The ice station on Floe 2 was set up on an aggregation of different ice types: refrozen leads, first year ice (FYI), and second
year ice (SYI). Modal sea ice thickness at the field station was 0.3 m, 0.9 m, and 1.7 m for refrozen leads, FYI, and SYI,
respectively (Rösel et al., 2017). Snow depth was on average $0.56 \pm 0.17$ m on FYI and SYI (Rösel et al., 2017), while on
refrozen leads it was approximately 0.02 m, likely redistributed from blowing snow. For this study, a 400 m × 60 m survey
field was established. Red flag poles, with black snow-filled trash bags, marked the outline making it visible from air (see
Figure 1). Shortly after OIB overflights, snow depth and sea ice thickness observations were collected on this two-dimensional
(2-D) survey field using a "snake line" sampling pattern with 5 m spacing between lines across the short axis of the field (see
Figure 1).

**2.2 Ground-based measurements**

Snow depth measurements ($hs_{SP}$, N=1046) were obtained with a GPS snow probe (SP) from Snow-Hydro (Fairbanks, AK,
USA). The snow probe is a thin pole with a sliding disk, 0.2 m in diameter. The pole penetrates the snow pack to the snow/sea
ice interface, while the disk rests on the snow surface. Inside the pole a magnetic device measures the distance between the
disk and the lower tip of the pole providing the snow depth (Sturm and Holmgren, 1999; 2018). Each measurement is time-
tagged and geolocated, and recorded on a data logger. The accuracy of the measurements over sea ice may vary between ±1-
3 mm (Marshall et al., 2006; Sturm and Holmgren, 2018) and the footprint is the size of the disk (i.e., 0.2 m). Snow depth
measurements were made approximately every 5 m following the snake line sampling pattern within the 2-D survey field (see
Figure 1).

Total snow and ice thickness measurements ($h_t$, N=7005) were obtained using the EM31 electromagnetic device (Geonics
Ltd., Mississauga, Ontario, Canada). A person dragging the EM31 instrument on a plastic sledge followed the snow probe
sampler. The EM31 measurements were sampled with a frequency of 2 Hz. The footprint size of the EM31 ranges from 3 m
to 5 m (e.g., Haas et al., 1997), depending on the ice and snow depth. The accuracy of the EM31 measurements is approximately
±0.1 m for level ice and decreasing over deformed ice (Haas et al., 2009).

For comparison to the EM31 data and to collect direct measurements, we drilled 10 equispaced holes around the 2-D field
perimeter with a 2" auger to measure ice thickness, snow depth, and ice freeboard. Ice thickness observations were made with
a thickness gauge from Kovacs Enterprises (Roseburg, OR, USA). The gauge is a specific tape measure with a foldable metal
weight at the bottom that can be deployed through the drill-holes. The accuracy of the readings is estimated to be ±0.01 m. In
addition, both, a snow pit was dug in the vicinity of the 2-D survey field (Merkouriadi et al., 2017c) to assess snow structure,
and an ice core was obtained to measure ice salinity, temperature, and density (Gerland et al., 2017). The core was extracted
with a 0.09 m diameter ice corer from Kovacs Enterprises (Roseburg, OR, USA).

To provide a regional context for the observations made in the 2-D field, we use a set of long, and independent, transects with
combined EM31 and snow depth measurements (N=5060) obtained within a maximum radius of 5 km around the ship during
the N-ICE2015 expedition. These were performed to characterize the spatial variability of snow and ice thickness in the area
surrounding the main ice camp. Further details can be found in Rösel et al. (2018). We use the 2-D grid snow depth
measurements and those sampled via transects within a 5 km radius, to provide spatial representativeness and context from
local- to regional-scales.

**2.3 Airborne measurements**

The OIB aircraft surveyed the 2-D survey field three times (see Figure 2) on 19 March 2015. First a surveillance overflight
occurred at 15:28 UTC. A second and third pass directly over the 2-D survey field occurred at 15:37 UTC and 15:43 UTC,
respectively. Because the first pass did not adequately intersect the 2-D survey field, we focus our analysis on measurements
obtained during passes 2 and 3 of the aircraft. Although the ice floe drifted during the airborne survey, the alignment of
transects 2 and 3 were such that they directly intersected the 2-D survey field on both passes.

For sea ice studies, the aircraft was equipped with an ATM laser altimeter system (Krabill et al., 2002), an ultra-wideband
frequency modulated continuous waveform (FMCW) snow radar system (e.g., Yan et al., 2017) and a digital camera system
(DMS) that provides high-resolution (0.1 m) geolocated visible-band images of the snow surface (Dominguez, 2018) allowing
for visual interpretation of sea ice conditions in the vicinity of the 2-D survey field (see Figure 1).

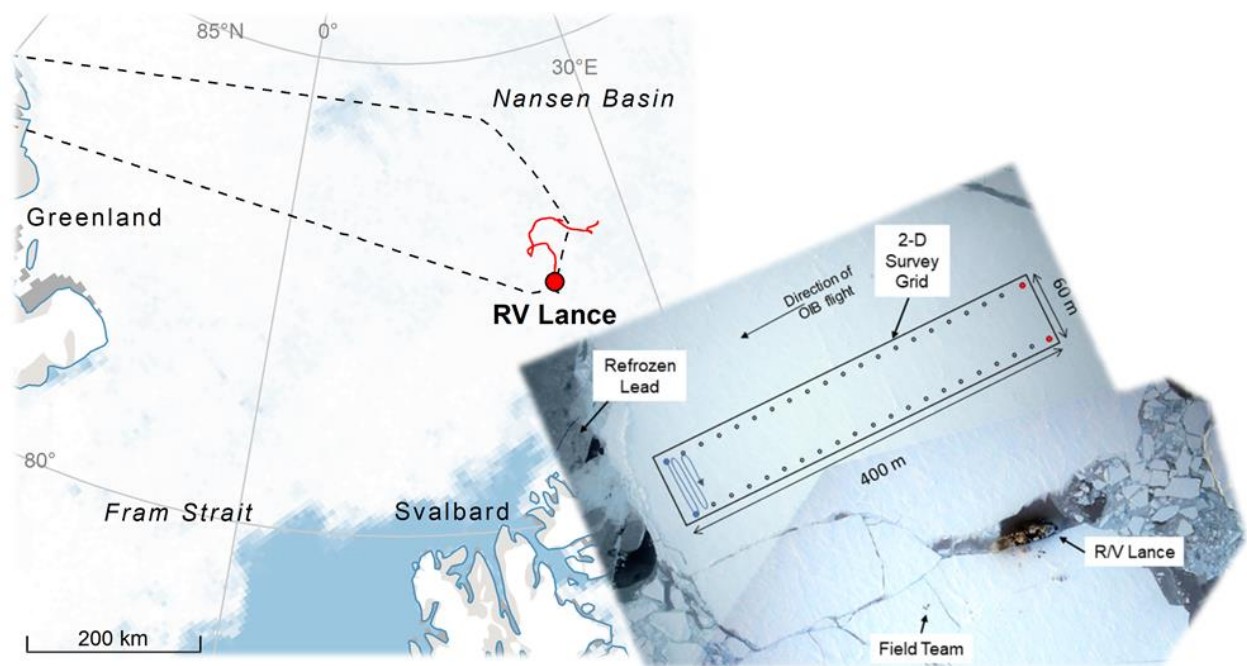

**Figure 1.** Overview of the location in the Arctic Ocean (left) and setup of the 2-D in-situ survey field situated on an ice floe as a part of the *floe 2* drifting phase to the west of *R/V Lance* on March 19, 2015 (right). Digital Mapping System (DMS) imagery (Dominguez, 2010, updated 2018) acquired during the OIB overflights were mosaicked to produce the aerial overview map. Black dots indicate the outline of the 2-D survey field. Snow and ice thickness measurements were obtained along the snake-line sampling pattern, as indicated in the left of the survey field.

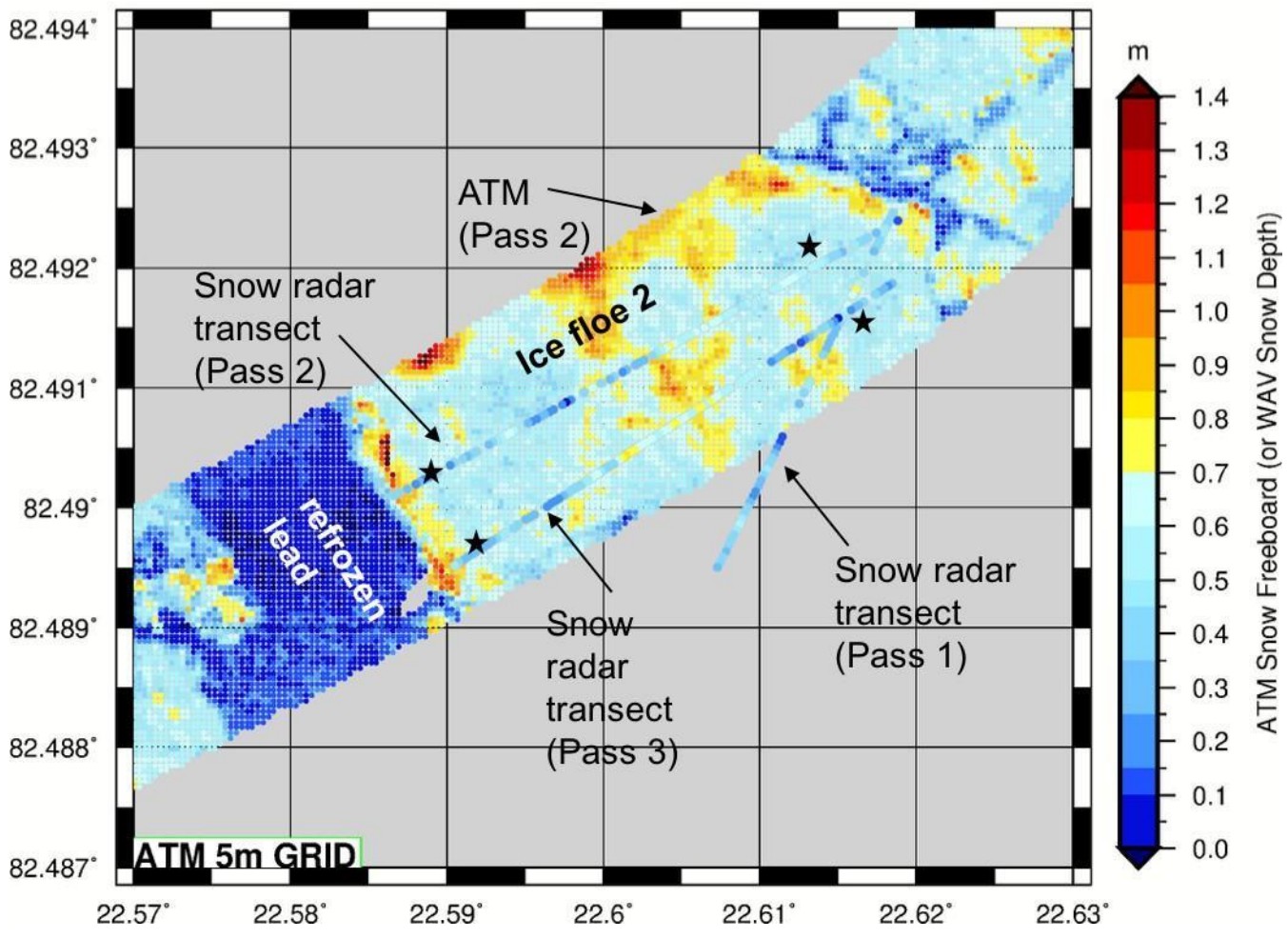

**Figure 2.** Detailed airborne mapping of the snow freeboard (5 m grid, derived from ATM observations of surface elevation) and snow depth (superimposed dots, derived from the airborne snow radar) at the 2-D survey field (corner points indicated by black stars) located on *Floe 2*. The three airborne transects across the field are indicated. During the OIB survey, the ice floe drifted south at an approximate drift speed of 0.15 ms$^{-1}$. WAV Snow depth in the secondary y-axis refers to the snow depth retrieved using the NOAA Wavelet technique (Newman et al., 2014).

173

**2.4 Methodology**

Table 1 summarizes all the variables used in the following context.

| suggested name | what it means |
|---|---|
| **ht** | total (snow + sea ice) thickness |
| **ht$_{EM}$** | total (snow + sea ice) thickness measured by EM |
| **Total (snow + ice) freeboard (also: snow freeboard)** | |
| hfbs | Total freeboard generally |
| hfbs$_{IS}$ | ...from drill-holes (IS for in-situ) |
| hfbs$_{ATM}$ | ...from laser scanner (ATM) |
| **Sea ice thickness** | |
| hi | Sea ice thickness generally |
| hi$_{IS}$ | ...from drill-holes (IS for in-situ) |
| hi$_{EM,SP}$ | ...estimated from EM and snow probe |
| hi$_{ATM,SP}$ | ...from ATM total freeboard, snow probe depths and densities |
| hi$_{ATM,SR}$ | ...from ATM and snow radar on the 2D-survey field |
| hi$_{ATM,SR(all)}$ | ... from ATM and snow radar data in a 10 km radius around *RV Lance* |
| **Snow depth** | |
| hs | Snow depth generally |
| hs$_{IS}$ | ...from drill-holes or snow pits (IS for in-situ) |
| hs$_{SP}$ | ...from snow probes |
| hs$_{SR}$ | ...from snow radar |

| Ice freeboard | |
|---|---|
| hfb | Ice freeboard generally |
| $hfb_{IS}$ | ...from drill-holes (IS for in-situ) |
| $hfb_{ATM,SR}$ | ...from ATM and snow radar |
| $hfb_{EM,SP}$ | ...estimated from EM and snow probes |
| $hfb_{ATM,SP}$ | ...estimated from ATM and snow probes |

**Table 1:** Overview of variables used in this study

### 2.4.1 Drift correction

To obtain spatial coincidence between the in-situ and airborne measurements of snow depth, freeboard and sea ice thickness, the positions of all measurements were corrected to mitigate the impact of the drifting sea ice during the experiment. As a reference, we determined the position of the four corners of the 2-D survey field using the DMS imagery collected during the second and third OIB overpasses. By comparing the differences for each corner marker between the two overpasses, we were able to deduce that the ice floe was drifting south at a speed of 0.15 ms$^{-1}$. To correct for the drift that occurred during the EM31 and SP sampling of the 2-D survey field, we followed the procedure described in Rösel et al. (2018): The EM31 data was resampled onto the coordinates of the SP track, and a Gaussian filter was applied to the EM31 data. Afterwards, both the EM31 and the SP data were interpolated on a 5 m regular grid.

### 2.4.2 Density of sea water, ice and snow

In all calculations we used the following values: the density for sea water was $\rho_W$ = 1027 kgm$^{-3}$ (Meyer et al., 2017), the bulk density for the snow pack was $\rho_s$ = 328 kgm$^{-3}$ (Merkouriadi et al., 2017b), and the bulk density for sea ice was $\rho_i$ = 910 kgm$^{-3}$ (Gerland et al., 2017). All values are based on measurements obtained during the N-ICE2015 expedition at floe 2.

### 2.4.3 In-situ snow depth sea ice thickness, and freeboards

In Figure 3, the concept of isostatic equilibrium is shown for four cases: On the left side, the ratio of snow depth ($h_s$) to sea ice thickness ($h_i$) is smaller, resulting in a positive sea ice freeboard ($hfb$) or ice freeboard at sea level. On the right side, the situation of the 'new' Arctic as described in Rösel et al. (2018) is schematically presented: A thick snow layer $h_s$ is pushing a

relatively thin sea ice $h_i$ layer below the ocean surface. The resulting sea ice freeboard $hfb$ becomes negative, and subsequently
the sea ice surface is vulnerable to flooding.
To obtain sea ice thickness ($hi_{EM,SP}$) from SP and EM31 measurements, the resampled snow depth measurements from SP
($hs_{SP}$ , N=1046) were subtracted from the total sea ice thickness from EM31 measurements ($ht_{EM}$ , N=1046):

$$hi_{EM.SP} = ht_{EM} - hs_{SP} \tag{1}$$


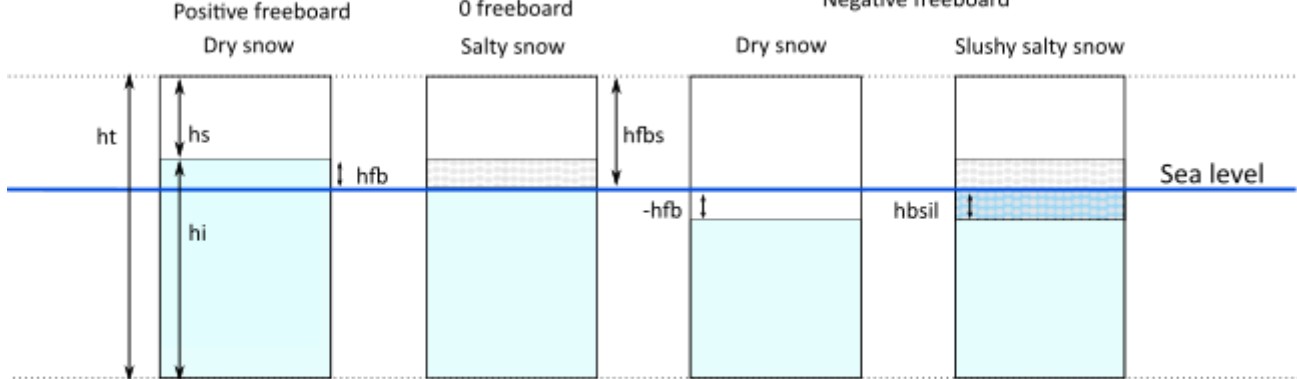


**Figure 3**: Some examples to show the concept of isostatic equilibrium of sea ice: On the left (a), the ratio of snow depth ($h_s$)
to sea ice thickness ($h_i$) is small, sea ice freeboard ($hfb$) is positive. On the right (c, d), the ratio of $h_s$ to $h_i$ is high, $hfb$ is
negative. In the second example (b), while the sea ice freeboard ($hfb$) is zero, the lower part of the snow pack can be salty
from brine wicking. This can also occur for positive ice freeboards. Example d) shows a slushy salty snow layer ($hfbsil$) due
to surface flooding, whereas example c) has a dry, non-salty snow cover. The snow freeboard $hfbs$ is the same in all four
cases.

Assuming isostatic equilibrium assumption, $hi_{EM,SP}$ under dry snow conditions can be calculated as:

$$hi_{EM,SP} = hfbs_{EM,SP} \left( \frac{\rho_W}{\rho_W - \rho_i} \right) - \left( \frac{\rho_W - \rho_s}{\rho_W - \rho_i} \right) hs_{SP} \tag{2}$$

which results in the freeboard ($hfbs_{EM,SP}$) derived from the snow probe (SP) and electromagnetic measurements (EM) using
obtained snow depth and sea ice thickness information and densities given above:

$hfbs_{EM,SP} = \frac{hi_{EM,SP}(\rho_W - \rho_i) + (\rho_W - \rho_S)hs_{SP}}{\rho_W}$     (3)

For wet snow conditions, or a flooded state of the sea ice, we refer to the studies of Zwally et al. (2008) and Ozsoy-Cicek et
al. (2013), where either $hs$ is set equal to $hfbs$, or a slush layer is included in the calculations, respectively. In addition, we
gain in-situ information from the drill-hole readings: Sea ice thickness ($hi$), snow depth ($hs$), freeboard ($hfb$), and snow
freeboard ($hfbs$)

$hfbs = hfb + hs$     (4)

As described in Rösel et al., 2018 the uncertainty of the ice freeboard $hfb_{EM,SP}$ and the total freeboard $hfbs$, resulting from the
propagation of uncertainties in the snow and ice densities and the sampling uncertainty, is estimated to be on average ±0.06
m. The accuracy of freeboards $hfb_{IS}$ and $hfbs_{IS}$ from the in-situ drill-hole measurements is ±0.01 m (Rösel et al., 2018).
**2.4.4 Airborne snow depth, sea ice thickness, and freeboards**
The DMS images were used to identify the geographical coordinates of areas of open water (with little or no ice cover) within
the large refrozen lead, located in the southwest of the 2-D survey field site and adjacent to it (see Figure 1). ATM elevation
measurements associated with these areas were averaged to estimate the local sea surface height (SSH). The SSH within the
lead was then subtracted from all ATM elevations, to obtain the ATM snow freeboard ($hfbs_{ATM}$). Individual ATM
measurements were resampled on the same 5-m regular grid as the in-situ snow and ice measurements across the sea ice floe
(see Figure 2). The snow radar echoes from passes 2 and 3 from the OIB survey also illustrate the presence of open water,
refrozen leads and areas with deep snow cover on the N-ICE2015 ice floe (see Figure 4).

We calculated snow depth from snow radar ($hs_{SR}$), following the methodology of Newman et al. (2014). Since the basal snow
layers were saline in some locations, the snow/ice interface could not always be detected. Therefore, a running average at 25
m length-scale (equivalent to five snow radar measurements) was used to account for an observed diffuse snow/ice interface
at the 2-D survey field site, possibly caused by a saline basal layer in the lower snow pack.

Ice freeboard ($hfb_{ATM,SP}$) and sea ice thickness ($hi_{ATM,SP}$), including a potentially refrozen slush layer, can be derived from a
combination of the airborne data measurements acquired over the 2-D survey field site with the in-situ snow-probe data and
were calculated as follows:

$hfb_{ATM,SP} = hfbs_{ATM} - hs_{SP}$     (5)

$$hi_{ATM,SP} = \frac{\rho_W hfb_{ATM,SP}}{\rho_W - \rho_i} + \frac{\rho_s hs_{SP}}{\rho_W - \rho_i} \qquad (6)$$

In addition, ice freeboard can be calculated through the difference between the ATM snow freeboard ($hfbs_{ATM}$) and the snow
radar snow depth ($hs_{SR}$). $hfb_{ATM,SR}$ is effectively the freeboard of a radar reflecting layer, including the ice freeboard plus a
frozen snow-ice basal layer, if present.

$$hfb_{ATM,SR} = hfb + hbsil + E = hfbs_{ATM} - hs_{SR} \qquad (7)$$

Where $hfb$ is the ice freeboard, $hbsil$ is the thickness of the slushy snow-ice basal layer, and $E$ is any remaining errors due to
the interface picking algorithms as applied to the snow radar echos (Figure 4).

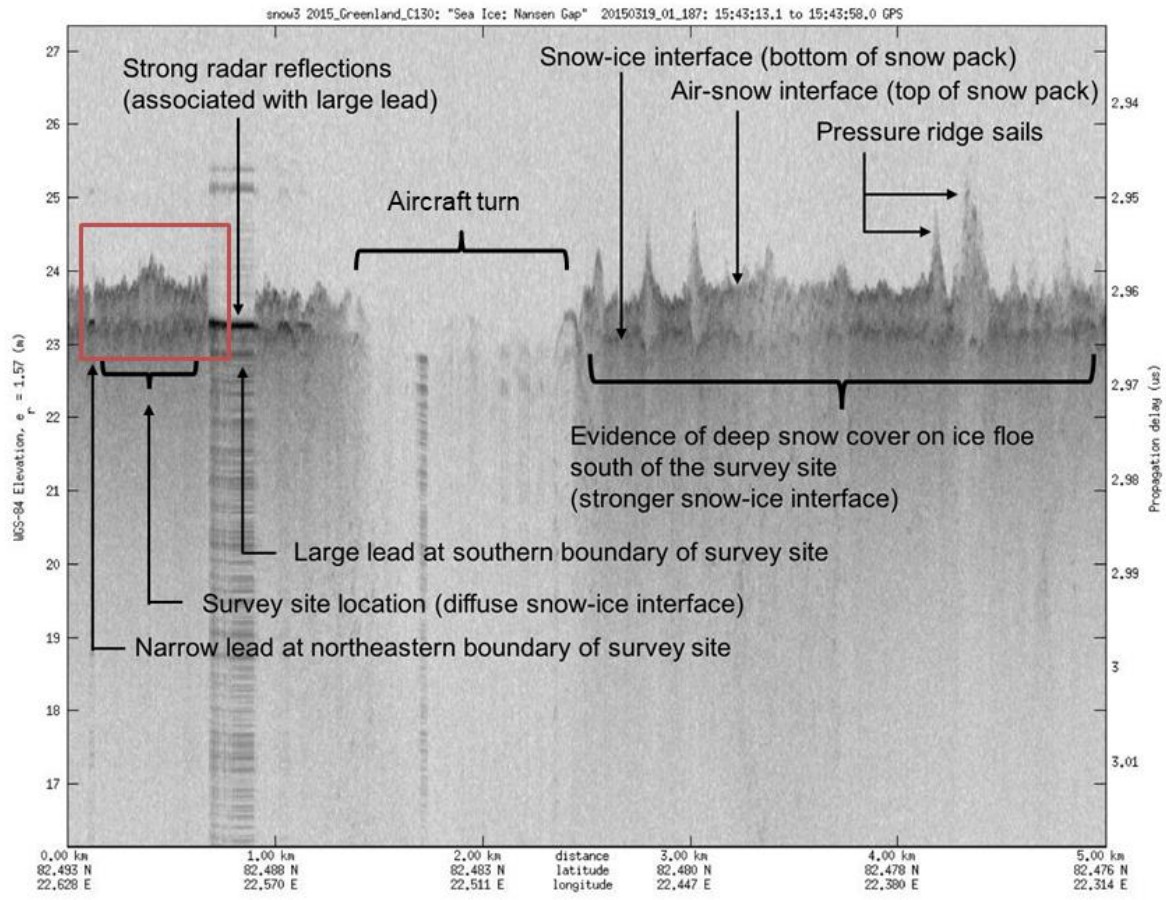



**Figure 4**: Processed and annotated OIB snow radar echo surveyed from the 2-D survey field, during the third pass on 19 March
2015 at 15:43 UTC. The red bounding box indicates the close-up of the region as shown in Figures 1 and 2.

**3 Results**
**3.1 In-situ and airborne measurements from the 2-D survey field site and their comparison**
The average calculated sea-ice thicknesses ($hi_{IS}$, Eq. 1) on the 2-D survey field site is $1.50 \pm 0.28$ m with a mode of 1.40 m,
and the average snow depth measured with the snow probe is $hs_{SP} = 0.58 \pm 0.15$ m with a mode of 0.55 m (results summarized
in Table 2). The drill-hole measurements lie within the standard deviation of all measurements collected at the 2-D survey
field site, i.e. our results demonstrate very good agreement across all observation methods (Figure 5 and Table S1 in
Supplement). Three out of the ten drill-holes were found to be flooded.
For direct comparison of the in-situ sampled snow depth and ice thickness data, a subset of the snow radar data of both
overpasses over the 2-D survey field site, limited by the 4 corner coordinates of the 2-D survey field (N=62), results
in an average snow depth of $hs_{SR} = 0.42 \pm 0.16$ m, with a mode of 0.40 m, 0.16 m and 0.15 m lower than the mean and
modal snow depth measured in-situ at the 2-D survey field site, respectively (N=1046, Figure 5a). However, the standard
deviations, i.e. the width and shape of the snow depth distributions, for both the in-situ and airborne snow radar observations
are in very good agreement with values of 0.15 m and 0.16 m, respectively. In addition, the average snow depth from the
airborne snow radar was 0.08 m smaller than average snow depth at the drill-hole locations $hs_{IS} = 0.50 \pm 0.18$ m (N=10, Figure
5a).

### 3.2 Local- vs regional-scale snow depth and sea ice thickness measurements

During the N-ICE2015 expedition, long transects on different predefined lines with combined EM31 and snow depth
measurements were performed to examine the spatial variability of the area surrounding the main ice camp and included
measurements of thin ice and deformed ice areas (Rösel et al., 2018). Altogether, five transects with 5060 gridded snow and
ice measurements were made on *Floe 2*, covering a time period from 24 February to 19 March 2015, which resulted in an
average snow depth of $0.55 \pm 0.18$ m and an average sea ice thickness of $1.09 \pm 0.92$ m. As stated in Rösel et al. (2018), the
snow and ice conditions were in average stable and did not change during the time of the drift.
As shown in Rösel et al. (2017), the overall measurements on the local area scale are representative of sea ice in the region.
To gain knowledge about the agreement in the snow depth between the airborne and in-situ observations on a more regional
scale, we compared observations from the OIB snow radar measurements from the same flight within a 10 km radius around
the position of *R/V Lance* with the average in-situ snow depth transect measurements during the drift of *Floe 2*. Similar to the
results obtained at the 2-D survey field site, the snow distributions show an offset for the airborne snow radar data towards
lower snow depth values. The average snow depth from the airborne snow radar was $0.49 \pm 0.25$ m, 0.06 m below the average
snow depth of $0.55 \pm 0.18$ m measured directly with the SP (Figure 5b). While the one-to-one comparison over the survey
field can be considered as a direct validation study, the statistical regional comparison across the larger area can potentially be
influenced by geophysical and thermodynamic processes such as ice dynamics, snow redistribution, snow metamorphism etc.
that occurred during the entire drift duration of *floe 2* (23 days) where in-situ data were acquired.

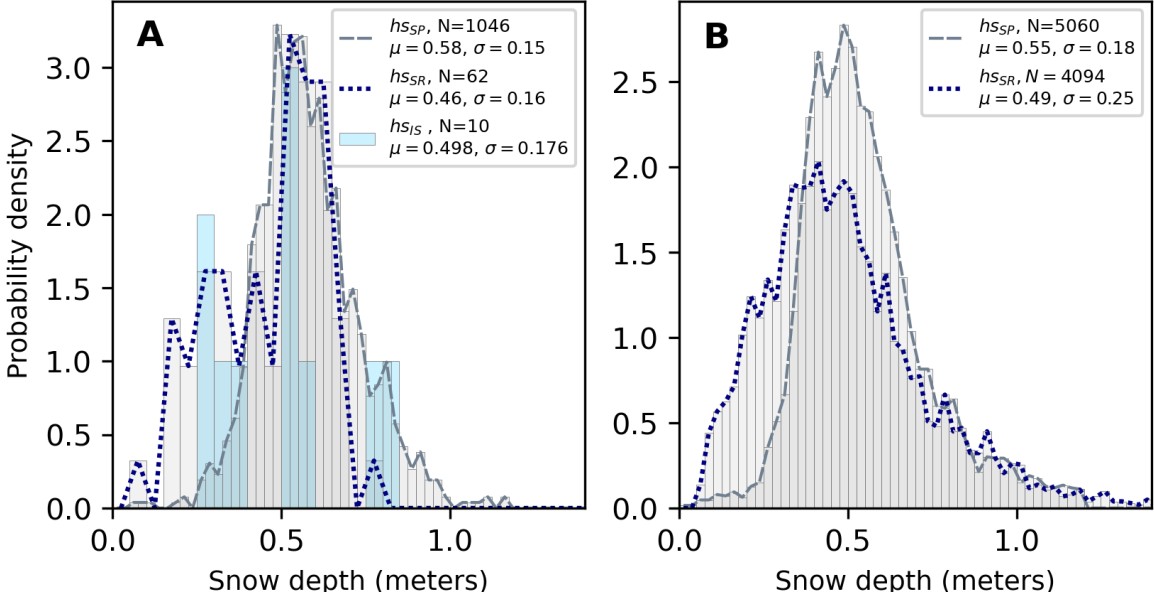

Figure 5. Probability density functions of snow depth measurements with given average values ($\mu$), standard deviations ($\sigma$), and number of measurements (N) from a) the in 2-D survey field site obtained with the snow probe (grey dashes), the OIB snow radar (blue dots) and drill-holes (light blue bars); b) a wider surrounding from snow probe sampling during the entire N-ICE2015 - floe 2 campaign (grey dashes) and from OIB snow radar within a radius of 10 km around the position of R/V Lance (blue dots).

For comparison with the ice freeboard, $hfb_{IS}$ = 0.01 ± 0.07 m, observed at the drill-hole sites, we used the in-situ ground measurements, i.e. SP and EM31, to derive freeboard $hfb_{EM,SP}$= -0.02 ± 0.05 m with an uncertainty of ± 0.06 m (Rösel et al., 2018), following Eq. 3.

While the average freeboard at the 2-D survey field site is close to 0 m based on the drill-hole measurements alone, the distribution of freeboards shown in Figure 6 are negative, with magnitudes up to 0.1 m. Results in the same range are obtained by subtracting the snow probe measurements from ATM surface elevation, resulting in an average value of $hfb_{ATM,SP}$ = 0.03 ± 0.09 m (see Figure 6). Taking the ±0.06 m uncertainty into account, this results in a negative freeboard area fraction of 19% and 14% across the 2-D survey field site for $hfb_{EM,SP}$ and $hfb_{ATM,SP}$, respectively (see Figure 7). An estimate of the ice freeboard plus the thickness of the snow-ice basal layer at the survey site that was impacted by brine wicking may be obtained by subtracting the snow radar measurements from the nearest ATM surface elevation value, which results in an average

$hfb_{ATM,SR} = 0.20 \pm 0.10$ m, but varies across the site (Figure 7c). The subsequent difference between $hfb_{ATM,SR}$ and $hfb_{ATM,SP}$
provides an approximate estimate of the thickness of the flooded, slushy, snow-ice basal layer of the snow cover.


| | snow depth ($hs$) [m] | sea ice thickness ($hi$) [m] | snow freeboard ($hfbs$) [m] | sea ice freeboard ($hfb$) [m] |
|---|---|---|---|---|
| in-situ (EM31/SP) | 0.58±0.15 | 1.50±0.28 | 0.54±0.09 | -0.02±0.05 |
| in-situ (drill-holes) | 0.50±0.18 | 1.39±0.33 | 0.50±0.12 | 0.01±0.07 |
| OIB (Snow radar) | 0.42±0.16 | | | |
| OIB (ATM) | | | 0.62±0.10 | |
| OIB (ATM/in-situ) | | (1.52±0.57) | | 0.03±0.09 |


321          **Table 2**: Results of snow, sea ice, and freeboard measurements and calculations of the 2-D survey field site


Finally, in Figure 8, we show the sea ice thickness distributions collected at the 2-D survey field site $hi_{EM,SP}$ and $hi_{IS}$, as well
as for the region surrounding the *R/V Lance*, $hi_{ATM,SR(all)}$. For comparison we include $hi_{ATM,SP}$ calculated from a combination of
the ATM data and the in-situ snow probe measurements. The in-situ measurements show that the 2-D survey field site was
situated on an ice floe ranging between 1.4 and 1.5 m thick (Figure 8). This compares to a regional-scale thinner ice cover
($1.09 \pm 0.92$m), as measured from Floe 2. The variability in sea ice thickness in the region surrounding the *R/V Lance* is about
three times larger than that at the 2-D survey field site. This is to be explained with a higher variability of ice types that were
covered during the regular transects on floe 2, including as well thin ice areas. We note that the average sea ice thickness of
the 2D survey site, $hi_{ATM,SP} = 1.52$ m, is only slightly above $hi_{EM,SP} = 1.50$ m, although the thickness equation (Eq. 6) does not
take into account the two-layer snow set-up, with each snow layer having a different depth and density. This is consistent with
the result shown in Figure 6, which presents the same freeboards for $hfb_{EM,SP}$ and $hfb_{ATM,SP}$. Comparing the distributions of
$hfb_{ATM,SP}$ and $hfb_{EM,SP}$ with $hfb_{ATM,SR}$, (Figure 6), $hfb_{ATM,SR}$ has a clear bimodal distribution with a first mode at -0.05 m, which
indicates the main scattering horizon of the snow radar, and the second mode at 0.25 m. This high second mode is potentially
caused by wet or saline snow, pushing the main reflecting horizon for the snow radar upwards as it will be discussed below.



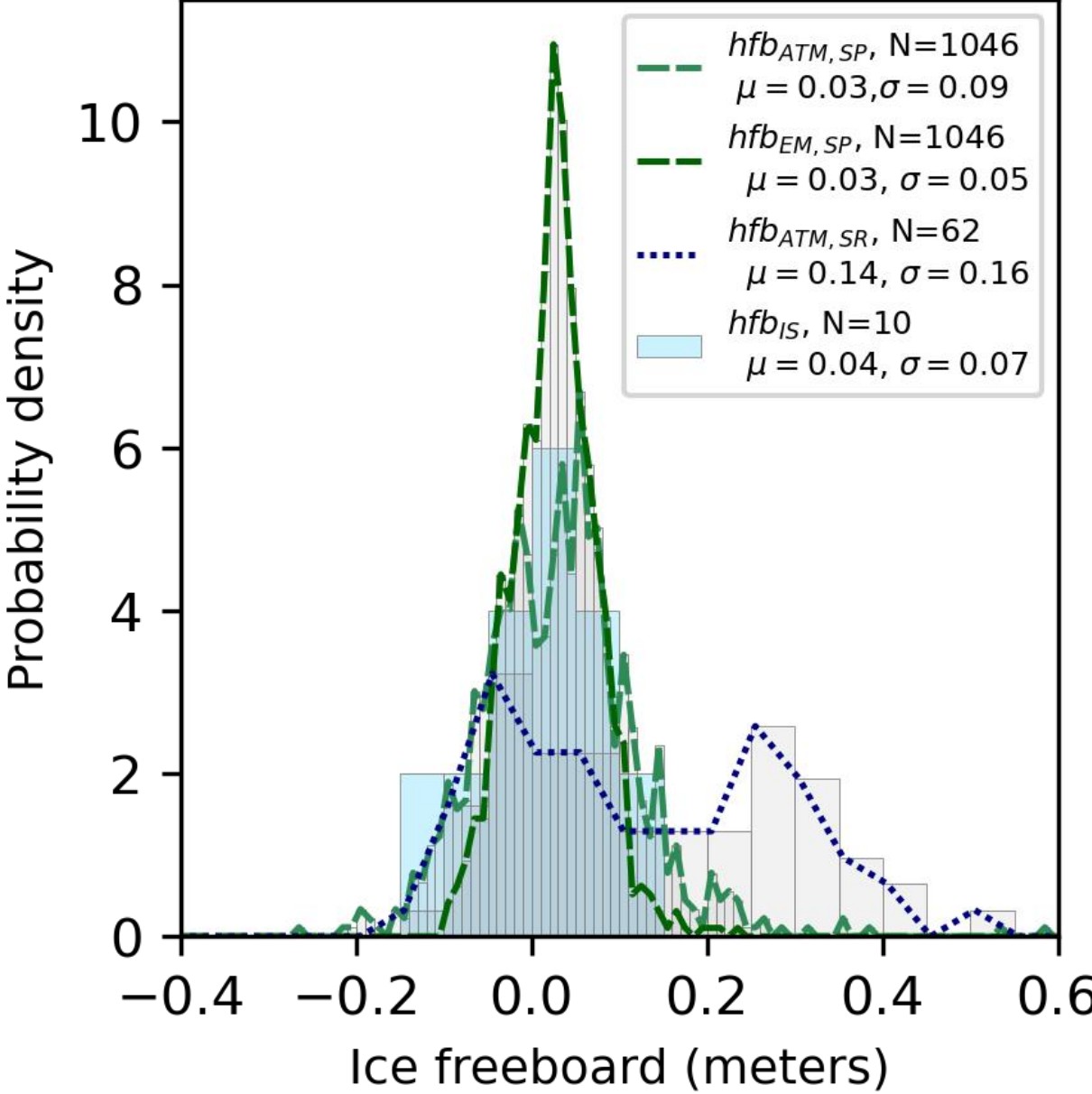


**Figure 6**: PDFs of ice freeboard (hi) with given average values ($\mu$), standard deviations ($\sigma$), and number of measurements (N)
from the 2-D survey field site. $hfb_{ATM,SP}$ (light green dashes): freeboard calculated from SP snow depth and ATM surface
elevations using Eq. 2; $hfb_{EM,SP}$ (dark green dashes): ice freeboard from EM and SP measurements; $hfb_{ATM,SR}$ (dark blue dots):
ice freeboard derived from ATM surface elevation minus matched SR snow depths within the 2-D field; $hfb_{IS}$ (light blue bars):
ice freeboard from drill-hole observations on 2-D field edges, plotted as a regular histogram for tidy visualisation.

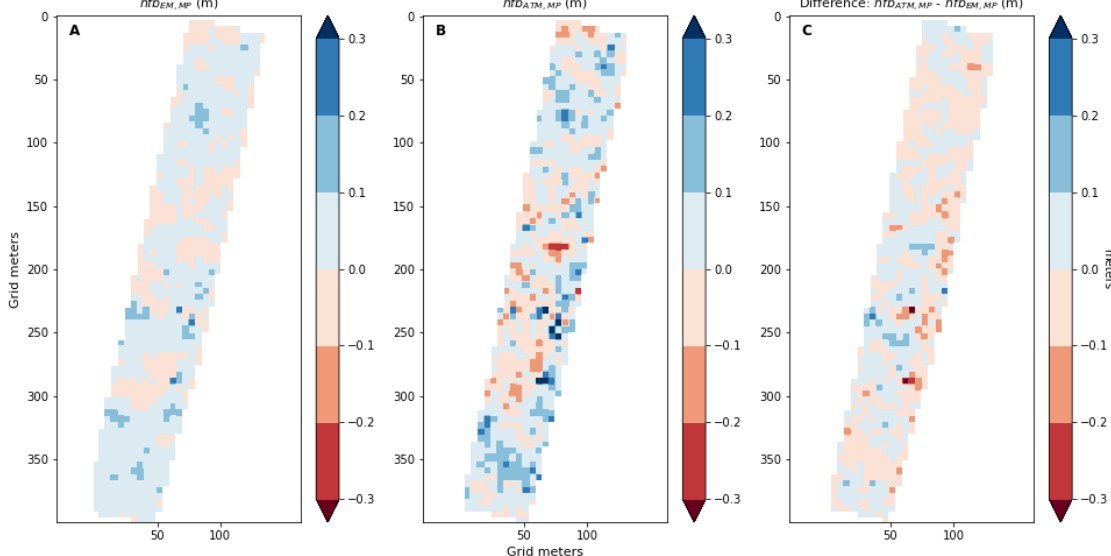


**Figure 7**: Ice freeboards ($hfb$) over the survey site. a) $hfb_{EM,SP}$, computed using Eq. 3 with EM and snow probe data gridded at
5 m. b) $hfb_{ATM,SP}$, using ATM surface elevation and SP snow depths gridded at 5 m. c) The difference between $hfb_{EM,SP}$ and
$hfb_{ATM,SP}$.







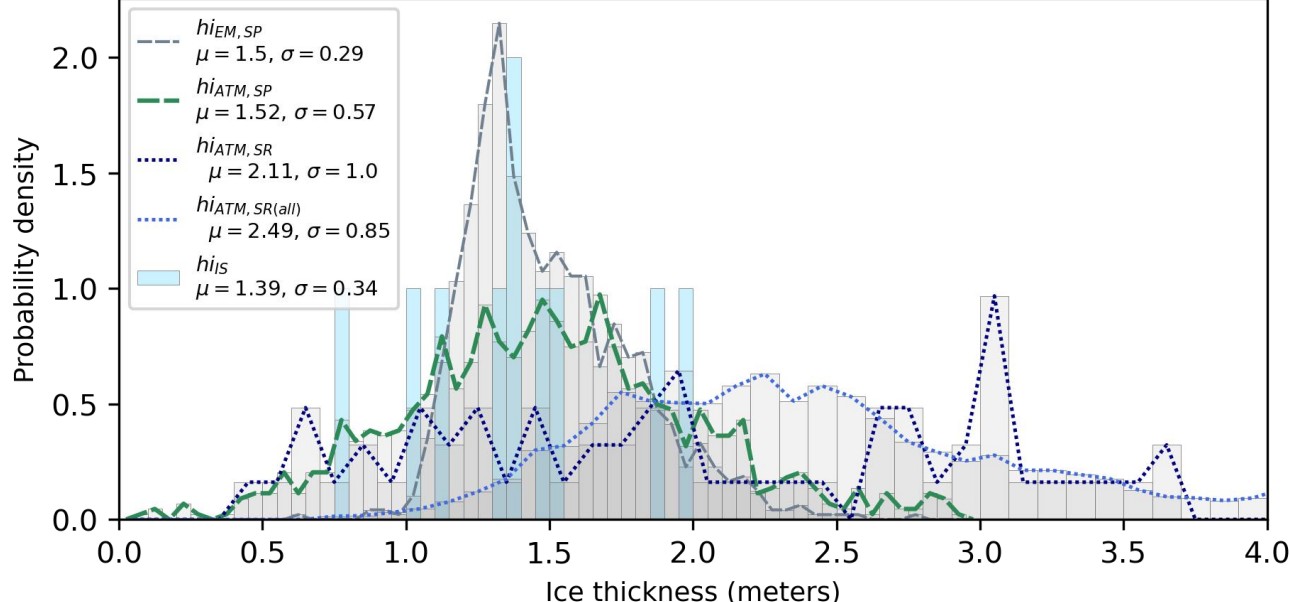

**Figure 8**: PDFs of sea ice thickness ($h_i$) with given average values ($\mu$), standard deviations ($\sigma$), and number of measurements (N) from the 2-D survey field site. $hi_{EM,SP}$ (dashed gray): ice thickness calculated from EM31 total thickness and SP snow depth; $hi_{ATM,SP}$ (dashed green): ice thick calculated from ATM surface elevation and SP snow depth in the 2D field; $hi_{ATM,SR}$ (dark blue dots): ice thickness calculated from ATM surface elevation and snow radar data using matched OIB ATM and radar observations in the 2D field; $hi_{ATM,SR(all)}$ (blue dots): ice thickness from all ATM surface elevation measurements in a 10 km radius around *RV Lance*, and the mean of all radar snow depth estimates; $hi_{IS}$ (light blue bars): ice thickness measured in-situ at drilling sites around the survey plot.

## 4 Discussion and Conclusions

The mean and modal snow depth estimates derived from the snow radar were 0.12 m lower than the in-situ snow probe measurements obtained at the 2-D survey field site. Over a larger regional scale of 10 km radius from the *R/V Lance* location, snow depth estimates derived from the snow radar underestimate in-situ snow probe-derived snow depth by 0.06 m, which is close to the measurement uncertainty of the snow radar system associated with its range resolution (Newman et al., 2014). In radar altimetry, it is assumed that the radar signal penetrates completely through a dry snow pack and energy is reflected from the snow/ice interface, which represents the height of the sea ice-freeboard above local sea level. This assumption is valid

(Beaven et al., 1995) for a cold, dry and homogenous snow pack, typical of Arctic sea ice in winter. However, for snow packs exhibiting high moisture content or higher densities (e.g. due to ice lenses/crusts), radar signals undergo absorption within the snow volume (e.g., Kwok and Maksym, 2014; Ricker et al., 2015). Recent studies suggest reduced signal penetration into the snow pack, with a more diffuse snow/ice interface, on both Arctic and Antarctic sea ice (Willatt et al., 2010, 2011, Gerland et al., 2013; Kwok and Kacimi, 2018), especially if the snow pack is saline (Nandan et al., 2020; Nandan et al., 2017) or very deep with ice lenses present (King et al., 2018). In addition, deep snow pushes the ice surface below the water level, leading to negative freeboard and might induce flooding and formation of highly-saline slush layers in the basal layers of the snow pack, which, when measured with a radar altimeter system, can result in a dominant scattering horizon above the true snow/ice interface (Nandan et al., 2020), and hence an overestimation of ice freeboard and thus sea ice thickness, and an underestimation of snow depth (Figure 6 and 8).

On FYI, overlying snow also wicks brine upwards from the sea ice surface during freeze-up, producing saline snow layers, predominately observed in the bottom-most 0.06-0.08 m of the snow pack (Drinkwater and Crocker, 1988; Geldsetzer et al., 2009; Nandan et al., 2016, 2017, 2020). The salinity profile of the ice core, taken on 5 March 2015 at the vicinity of the 2-D survey field site, shows a typical C-shape profile with relatively high salinity values up to 11.3 psu at the top and 5.8 psu at the bottom, respectively, and lower values of between 1.3 psu and 4.3 psu in the middle sections of the ice core (see Figure 9) suggesting that the 2-D survey field site on the ice floe comprised of FYI. Snow salinity observations from snow covers (0.26 m and 0.34 m thick) overlying the FYI floe indicate highly-saline, 0.10 m deep basal layers in both snow covers, by up to 10 psu. Additionally, one-third of the drill-holes at the 2D field site indicated flooding of the snow pack, and negative freeboard, which induced the formation of highly saline and saturated slush in the basal snow layers. Presence of slush layers at the 2D field site resulted in a challenging geophysical setting for the measurement of snow and ice thickness using remote sensing techniques, which involve snow radar measurements.

Previous studies (Barber et al., 1998; Barber and Nghiem, 1999; Nghiem et al., 1995; Geldsetzer et al., 2009; Nandan et al., 2017; Nandan et al., 2020) have reported the impact of saline snow on FYI, which alters the geophysical, thermodynamic, dielectric and radar scattering properties of the snow cover, thereby impacting radar signal penetration through the snow pack. Nandan et al. (2017) showed that a saline snow cover on a positive freeboard, landfast FYI setting, induced by upward snow brine wicking from the sea ice surface, shifted the main radar scattering horizon away from the snow/ice interface by up to 0.07 m. In these studies, covering the Canadian (Nandan et al., 2017) and the Atlantic (Nandan et al., 2020) sectors of the Arctic, the conditions at the survey field site included saline, wet, and deep snow which impacted the accuracy of the snow depth derived from the snow radar signal. In the Nandan et al. (2020) case study from the N-ICE2015 experiment, they demonstrated significant overestimations in FYI thickness by up to 95% between simulated FYI thickness, snow radar and ATM-derived FYI thickness. They simulated the Ku-band radar scattering horizon from 0.36 m and 0.45 m deep snow on 0.69 m and 0.92 m thick ice, exhibiting negative freeboards by 0.04 m and 0.07 m, respectively. Measured snow salinities towards the basal layers overlying slush layers, were found to be high, up to 25 psu. They found that the FYI thickness overestimations

was a result of vertical shift in the radar scattering horizon, caused by upward snow brine wicking from the slush layers, caused
by negative freeboards.
Our study shows that saline snow conditions can lead to the observed underestimation of snow radar-derived snow depth. This
is likely due to a combination of factors including reflection from a scattering horizon in the snow pack that is above the main
snow/ice interface, a diffuse scattering horizon within the snow volume, and potential errors in the height of the snow/ice
interface picked in individual snow radar echoes. Although, we do not have any direct measurements of slush salinity, nor any
indication of whether the high basal snow salinity values observed from our survey site and also reported in Nandan et al.
(2020) are due to basal snow brine wicking from the slush layers. Even though the snow radar underestimated mean snow
depth by between 0.12 m and 0.06 m across the 2-D survey field site and the regional survey, the radar was able to fully
reproduce the snow depth variability, when compared to in-situ measurements (standard deviations of 0.16 m and 0.15 m for
the survey field, respectively; see Figure 3). Thus, we can report that the airborne snow radar is capable of measuring
meaningful snow depth distributions even in challenging snow pack conditions. However, ambiguous radar signal penetration
through slushy layers (caused by sea ice flooding) and saline snow covers (caused by brine wicking from sea ice surface) may
introduce a potential bias in accurate estimates of snow depth, and subsequently the resulting calculations on sea ice thickness
as shown in Figure 8. In our field experiment we can clearly see an overestimation of the sea thickness, calculated from ATM
surface elevation and snow radar data.
For a radar altimeter, the main scattering horizon within the snow volume is not just a function of snow depth, but also depends
on the thermodynamic properties of the snow cover (i.e., snow temperature, density, salinity, wetness, roughness, and grain
microstructure). Further research is required to understand the relationship between the main scattering horizon and variability
in snow cover properties. Besides of this, the different scales of high-resolution snow depth observations from the snow probe
versus the low-resolution snow radar measurements, as well as the different temporal resolution, especially of the regional
observations might have an effect on the bias of the snow radar measurements. Again, here further research might be necessary
to fully understand the complexity of the system.
Biases caused by uneven penetration of radar altimeter signals within slushy and saline layers in the snow pack will also have
implications on estimates of snow and sea ice thickness measurements from currently operational satellite-based radar
altimeters such as SARAL AltiKa (Ka-band), CryoSat-2 and Sentinel-3A/B (Ku-band), and the ESA's forthcoming Ku- and
Ka-band dual-frequency satellite radar altimeter mission CRISTAL. King et al. (2018) reported underestimation of sea ice
thickness derived from CryoSat-2 data caused by negative freeboards in the same region as was investigated in our study. A
detailed quantification of the contributions to the error budget associated with freeboard retrieval from CryoSat-2 was made
by the ESA CryoVal-SI project team and is described in Ricker et al. (2014) and Haas et al. (2016). To examine the impact of
a deep snow pack with saline and/or flooded snow/ice interface, we show as an example the monthly averaged CryoSat-2 sea
ice products provided by the NASA Goddard Space Flight Center (GSFC, Kurtz et al., 2014) for the region surrounding *R/V*
*Lance* location in March 2015 (Figure 10). Noticeably, the CryoSat-2 sea ice freeboard and derived sea ice thickness from this

region demonstrate large spatial variability. Freeboard measurements are up to 0.3 m (Figure 10b), and the derived sea ice thickness is overestimated by over 1.0 m (Figure 10c), compared with the in-situ results reported in Rösel et al., 2018). Modelled snow depths of 0.15 m and 0.37 m (derived from Warren et al., 1999; Kurtz et al., 2014, Figure 10a) are underestimated when compared to the observed in-situ snow depth, which averaged 0.55 m.

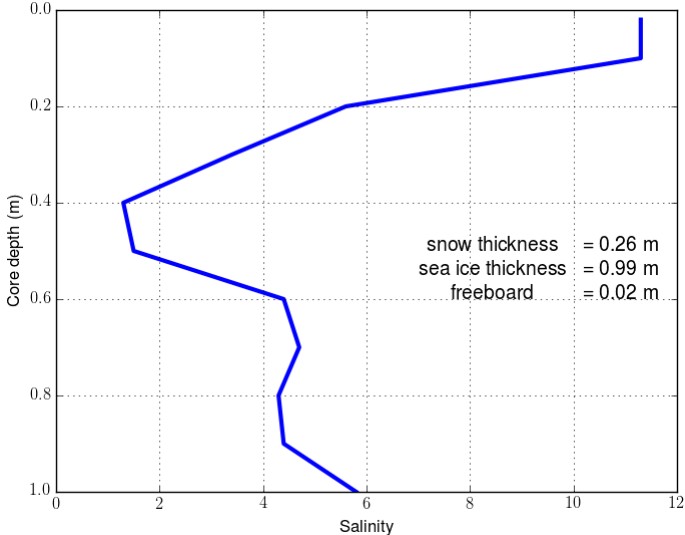

**Figure 9**. Salinity profile and auxiliary data of an ice core, taken on 5 March 2015 within the vicinity of the 2-D survey field.

Sea ice parameters from the GSFC CryoSat-2 are derived with a waveform fitting procedure using an empirical waveform model (Sallila et al., 2019), which should account for snow geophysical properties. However, presently operational CryoSat-2 retracker algorithms or empirical models (e.g. Hendricks et al., 2010; Ricker et al., 2014; Kurtz et al., 2014) do not account for snow pack flooding as a source of error, affecting the accuracy of sea ice freeboard and thickness estimates. Moreover, since our survey site was also drifting, we acknowledge the impact of sea ice dynamics also affecting the correlations between in-situ measurements and satellite-derived estimates, both acquired at different times (Tilling et al., 2018). All of these issues could cause misinterpretation of both airborne and satellite radar altimeter signals, especially in complicated areas where sea ice undergoes drift and frequent flooding of snow cover. These findings might have a minor impact for Arctic regions for now, where flooding of the sea ice is not as prominent as in Antarctica, but considering a changing Arctic snow and sea-ice regime, this might become a more prominent topic in the North as well. In order to obtain more accurate and realistic snow, ice, and freeboard measurements, we therefore recommend future improvements in sea ice freeboard and thickness retrieval algorithms.

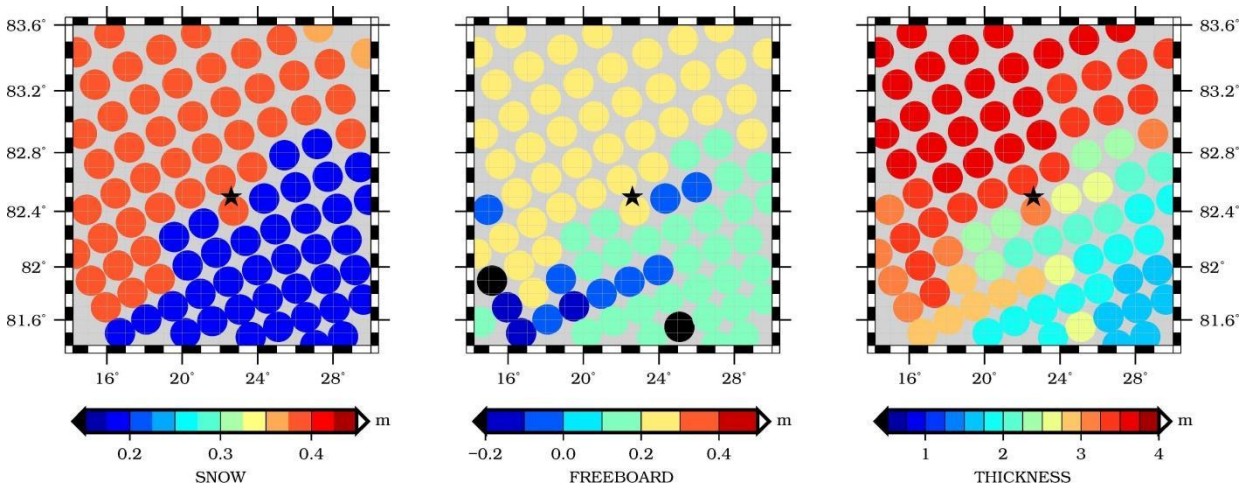

**Figure 10**: Results from CryoSat-2 sea ice products from GSFC, averaged for the month March 2015 for a region of 250 km × 250 km over the in-situ site in the Norwegian Arctic for a) snow depth, b) sea ice freeboard, c) sea ice thickness. The position of *R/V Lance* is marked with a star.

**Acknowledgements**: The authors would like to thank the two reviewers for their valuable comments which helped is to improve the original manuscript. We thank all crew members of *R/V Lance* and all scientific staff involved in N-ICE2015 for their support and help during the fieldwork. Figure 1 was created with the help of Anders Skoglund (NPI). This work was supported by the Norwegian Polar Institute's Center for Ice, Climate and Ecosystems (ICE) through the project N-ICE2015 and the project ID Arctic (Norwegian Ministries of Foreign Affairs and Climate and Environment; SG, AR). VN acknowledges Post-Doctoral Fellowship grant to Canada's Marine Environmental Observation, Prediction and Response Network (MEOPAR). SLF was supported under NOAA grant NA14NES4320003, and the NOAA Product Development, Readiness, and Application (PDRA)/Ocean Remote Sensing (ORS) program. GS acknowledges support by the Trans regional Collaborative Research Center (TR 172) "ArctiC Amplification: Climate Relevant Atmospheric and SurfaCe Processes,and Feedback Mechanisms (AC)[3]", funded by the German Research Foundation DFG. The contribution of AS to this study was funded by the Research Council of Norway through the Nansen Legacy project (NFR-276730). N-ICE2015 acknowledges the in-kind contributions provided by other national and international projects and participating institutions, through personnel, equipment and other support. N-ICE2015 data is available through http://data.npolar.no/ and NASA's OIB data is publicly available via https://nsidc.org/icebridge/portal/map.

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

**Supplement**

| Drill-hole label | Ice thickness | Snow depth | Freeboard | Flooded |
|:---:|:---:|:---:|:---:|:---:|
| D0 | 1.76 | 0.34 | 0.13 | no |
| D1 | 1.48 | 0.49 | 0.01 | no |
| D2 | 0.78 | 0.30 | 0.02 | no |
| D3 | 1.03 | 0.57 | -0.03 | no |
| D4 | 1.34 | 0.38 | 0.03 | no |
| D5 | 1.65 | 0.83 | -0.11 | yes |
| D6 | 1.35 | 0.76 | -0.09 | yes |
| D7 | 1.36 | 0.28 | 0.07 | no |
| D8 | 1.15 | 0.51 | -0.03 | yes |
| D9 | 1.96 | 0.51 | 0.07 | no |
| Mean (Stddev) | 1.39 (0.35) | 0.50 (0.19) | 0.01 (0.07) | |


**Table S1.** Measurements acquired at 10 drill-hole sites located randomly across the 2-D survey field site. All values are given
in meters.