# Peer review of "Implications of surface flooding on airborne estimates of snow depth"

_The Cryosphere, 2020_

## Referee Comment (RC1) · Anonymous Referee #1 · 17 Sep 2020

This paper presents an assessment of the impact of flooding on radar penetration in the snow layer over sea ice and its consequence on radar-based ice thickness estimates. This study represents a valuable contribution to the field of sea ice remote sensing as it relies on both airborne and in-situ measurements to quantify the bias due to snow-ice formation in retrievals. However, I have some comments that need to be addressed first.

Major comments:

-My main concern has to do with the writing of some sections of the paper. Indeed there are several confusing long sentences throughout the manuscript that challenge the overall understanding of the paper. I have to point out paragraph 2.2 (Ground-based measurements), which is really dense and needs to be re-written. I suggest that

the authors add sub-sections.

-The notation used for the different observation techniques and instruments is confusing. This results in equations using variables with too many subscripts (equation 6 for example). I suggest that the authors use only the name of the instruments in the definition of variables. This should greatly improve the quality of the paper.

Minor comments:

P2L39-42: the sentence" On the other hand . . .. melting" needs to be rephrased. Perhaps add parenthesis between "compared to" L40 and Perovic, 1996).

P2L41: Remove capital letter on "Snow reflects".

P2L56: Replace "snow/sea ice" by "snow/ice".

P3L75: Please specify which freeboard you are referring to (snow/ice).

P4L101: I suggest adding a link or reference to Snow-Hydro.

P11L265: Add a comma between "site" and "the snow distributions".

P13L289: Replace "0.00" by 0.

P14L305: Replace "asfor" by "as for"

P14L311: Avoid using "result" twice in the same sentence, I suggest to remove "the resulting".

P14L311-311: Please revise:" Our results suggest . . . two-layer snow situation".

P16, Figure 7: Please correct the legend: FB3 and FB4 are the same.

P17L345: replace "about the R/V" by "from the R/V".

P18L352: Seems to be an extra space between "undergo" and "absorption".

P18L364: Remove "of" in "of > 11 psu"

P18L371-372: Seems to be an extra space between "thereby" and "impacting".

P20:L415: replace "should to be" by "should be".

---

## Referee Comment (RC2) · Anonymous Referee #2 · 27 Oct 2020

Review of

Implications of surface flooding on airborne thickness measurements of snow on sea ice

by

Rösel, A., et al.

Summary: This is a very interesting study aiming for a better quantification of the limitations of radar (altimeter) measurements over snow-covered sea ice. Such measurements have been used since two decades to obtain an estimate of the Arctic sea-ice thickness by means of satellite sensors such as CryoSat-2 and since one decade to estimate snow depth on sea ice by means of airborne sensors such as OIB. The crux

of present-day radar (altimeter) measurements is the unknown penetration depth into a snow cover of unknown depth and physical properties. While dry cold snow is not problematic, deep snow with icy layers interspersed, or wet snow in all of its forms (from surface melting or from flooding at the ice-snow interface) can cause a significant bias in both, snow depth retrieval using the OIB snow radar and sea-ice freeboard and hence sea-ice thickness retrieval using a spaceborne radar altimeter. This study - based on a convincing set of contemporary observations obtained in-situ and from OIB sensors during the N-Ice2015 expedition - quantifies biases in the quantities required for spaceborne sea-ice thickness retrieval: sea-ice freeboard and snow depth, for deep snow (about 50 cm) on relatively thin (150 cm) sea ice with a considerable portion of the ice-snow interface being flooded. Since these conditions appear to be widespread in the Atlantic sector of the Arctic Ocean and the associated peripheral seas and are likely to increase in coverage in the entire Arctic Ocean in the future, the results are of high relevance for accurate retrieval of the sea-ice thickness using satellite radar altimetry.

Given the relevance of the results I recommend to improve the current manuscript along the lines suggested in the following general comments - which are further detailed in the specific comments.

GC1: The study shines through an excellent set of observations. There are parts of the description which ask for improvement, though. One is an improved consideration / discussion of snow depth and sea-ice thickness variations between survey site (local), 5 km circle around Lance (extended in-situ survey) and 10 km circle around Lance (OIB); more details regarding this issue I give in the specific comments. The discussion about scaling and representativity issues is light and could be expanded; elements of this topic I try to express with the following questions. Can one really, as done, combine the OIB data with the in-situ data with such high accuracy? Are EM31 total thickness measurements really that accurate that one can derive highly accurate sea-ice freeboard when combining this data with snow probe snow depth observations? How does

the EM31 signal respond to a possibly spatially extended area of flooded sea ice? How does the issue that OIB is known to underestimate thin snow / has issues over highly deformed sea ice influence your results? Finally, the one (or maybe two) ice salinity profiles used at the end to conclude that your observations prove that saline snow can be the main cause for the observed biases, are not very well connected to the rest of the manuscript - even though they are the dominant topic in the discussion and conclusion section. I recommend to give the description of these in situ observations more weight and demonstrate more clearly how well (hopefully) these single point measurements represent the conditions in the detailed survey area.

GC2: My impression is that the MAIN issue is that OIB data result in a sea-ice freeboard bias of 0.2 m. This is a SUBSTANTIAL bias but it is not presented and discussed in an overly prominent way in the manuscript. I recommend to be more exhaustive in the discussion of particularly these results in the context of Figures 6 through 8.

GC3: The secondary issue is the snow density of the proposed () two-layer snow setup which is discussed in the context of Figure 8, hi4. There are, to my opinion, at least two areas of improvement. The first one is superposing the drill-hole sea-ice freeboard data onto Figure 7 and discuss the results. The second one is to conduct a sensitivity study which plays around with possible snow layer depths and densities used in Equation 6 to derive hi4. A pre-requisite for these actions is an appropriate introduction of what you cann the two-layer snow setup, which is not adequately described yet in the manuscript. In this context, I also kindly ask for clarification of the wording "slushy basal snow layer" versus "snow-ice basal snow layer" because snow-ice is refrozen slush hence hard while slush is soft - which has implications for the snow probe measurements and the interpretation of the measurements.

Specific Comments:

Line 179++: I am not sure your choice of denoting different variables with 1, 2, 3, 4, using hi (I) and hs (S) and hfb (Fi) and hfbs (F) to name the variables, i.e. without

subscripts, and introducing subscripts such as EM, SP or IS is an optimal solution. In any case I recommend to use it in a consistent way. This means that also in the running text hi and hs should be given as used in the formulas (see Lines 174/175 for instance). In addition, I am wondering whether it is necessary to abbreviate "in situ" with IS. To me this is confusing in the zoo of short names. But this is of course your choice. If you keep IS then you need to move its definition from Line 194 to Line 179.

Equation 2: I am sure that the freeboard in this equation needs to be the snow freeboard, i.e. hfbs.; this is the classical equation to derive sea-ice thickness I from total (sea ice + snow) freeboard F and snow depth S, isn't it?

Then equation (3) would be the total freeboard as well and not the sea-ice freeboard. In order to end up with the sea-ice freeboard here I suggest to use the retrieval equation used for radar altimetry (in your notion): hi1IS = (hs1SP * rho_s + hfb1IS * rho_water) / (rho_water - rho_ice)

In Line 194 it needs to be "1" instead of "2" in the variable name.

Equation 4: In the context of this equation I'd like to note that you are not consistent with Figure 3. There you denote snow freeboard as hfbs while in Equation 4 and in Line 202 you write hsfb. I like hsfb more.

Line 206/207: While the accuracy for the drill-hole measurements is clear from the stated measurement accuracy of the measurement device, I am wondering whether you might want to add one sentence about the way you estimated the value of 0.06 m for the combination of snow probe and EM31 measurements.

Equation (7): I suggest to split this equation into two. Please first introduce hB without the interpretation that it can be decomposed into sea-ice freeboard, basal snow-layer thickness and an error term and subsequently, in a follow-on equation (if necessary) show the decomposition of hB. I note, after having read the entire manuscript, that there is limited reference to and usage of this equation. Or did I overlook a figure or

table where you provide an estimate of the basal snow-layer thickness?

Figure 4: I am a bit confused with what I see here. I appears to me that this plot starts on the left somewhere close to the top right corner of the map shown in Figure 2, taking one of the overpasses shown towards the southwest (crossing the survey area). Fine. Then comes the aircraft turn; the main part of the radar-echogram shown in Figure 4 is related to sea-ice conditions "on ice floe south of the survey site". Now, what I have problems with is the fact that you stated the times for overpass #2 and #3 as 15:37 and 15:43 in the caption of Figure 4 while in the heading line just above the figure it specifies a time range around 15:43, suggesting that this is only overpass #3. Could it hence be that we only look at overpass #3 and that only the small part (left quarter of the echogram) is coincident with the tracks (actually only overpass #3) denoted in Figure 2? If this is the case, I am wondering whether it wouldn't make sense to expand that left quarter of the echogram because this has a direct relation to the survey area.

Lines 253-255: "In addition ..." How important is it to explicitly mention the mean of these very few (compared to the other samples) snow depth observations at the drill hole sites in the text? Would it be sufficient to only show this value in the Table? I am suggesting so because you are coming up with quite a number of different snow depth values and it begins to become confusing. Particularly, since you repeat this information in Line 273.

Line 263: What is the motivation to draw a 10 km circle around R/V Lance for the air-borne data when the in-situ observations along the five transects were carried out within a 5 km radius circle?

Line 275/276: "Three ... before drilling." –> How do you know?

Figure 5: In the left panel you denote hs1 and hs3 with the respective methods while in the right panel you write "regional". My suggestion would be to be consistent. At this point the reader possibly knows that hs1 is based on snow probe data and that hs3 is based on the snow radar. Hence you could use "survey area" in the left panel and

keep "regional" in the right panel.

Lines 294-298: These sentences about brine wicking etc. fall a bit from heaven. I suggest to start a new paragraph here and motivate these further considerations by again stressing that near-zero ice-freeboard supports flooding of the ice-snow interface and subsequent upward wicking of seawater and/or brine into the basal snow cover. The paragraph about the c-profile of the salinity given at the beginning of Section 3 could be placed here in a much more logical way - as this piece of information seems a bit lost where it is located currently.

Figure 7: I suggest to overplot the freeboard values shown in Table 2 of the appendix onto the maps in each panel by using, e.g., a color-filled circle, and discuss what you see.

Lines 303-313 / Figure 8:

- Line 305: I don't find hi_REGIONAL in Figure 8.

- Line 307: "slightly thinner" –> It might be a matter of taste but a difference of 0.3 to 0.4 m in a thickness range between 1.1 and 1.5 m I would not call "slightly". In addition, those 5 regional survey lines, were these laid out without checking the ice conditions beforehand? What I mean by this is, that based on the large sea-ice thickness standard deviation it seems likely that these 5 lines had a substantial fraction of thin ice from re-frozen leads while the bulk of the sea ice inbetween might have had a similar thickness than the ice of the survey site. Please check and, if need be, re-phrase statements.

- I am missing commenting on hi3.

- Line 309-313: I suggest to re-write this part. hi4, if computed using Eq. 6, uses the sea-ice freeboard (hfb) which is derived from airborne ATM total freeboard (accurate) and the in-situ snow depth (accurate as well); in addition, for the snow part it uses (again) the in situ snow depth. The densities you used are those which you measured in the field = accurate. Hence, at first glance it is not clear why even with the densities

you measured the hi4 is biased compared to hi1 and hi2. The statement that the bias in sea-ice thickness between hi1 and hi4 of about 0.4 m is consistent with the bias in hfb of 0.03 m and that this is in part due to the not sufficiently well considered "two-layer snow set up" is not backed-up yet by your writing or the figures. If, as you suggest, snow densities across this two-layer snow set up are the main cause, then I strongly suggest to play around with potential snow densities (you measured some in the field, didn't you?) and layer thicknesses to figure out whether your hypothesis is true. In other words: Which snow density values of the proposed two-layer snow setup explain the observed difference between hi1 and hi4? Are these realistic?

Discussion section:

General comments: I don't see a reason why to write "total snow depth" or "total snow thickness". I guess "total" can be omitted.

Line 345: This is a reminder that it might make sense to provide a scientific motivation for using a 10 km radius for the airborne data versus a 5 km radius for the in situ data. If by chance the airborne data observed a comparably large fraction of thin ice, then the airborne snow depth would naturally be smaller and hence its bias to the in situ snow depth.

Line 357: "slushy, snow-ice formation in the basal layers of the snow pack" vs. Line 367: "formation of highly saline and saturated slush in the basal snow layers" –> It is a difference whether one speaks of wet / saturated slushy snow ... which is a rather soft material, or whether one speaks of snow-ice which - at least to my understanding - is refrozen ... and hence hard. It is, to my opinion, important to distinguish between those because I'd think that the SP measurements would penetrate slush and hence INCLUDE the thickness of the slush layer into the snow depth reading while these would not penetrate snow ice and exclude that part from the snow depth reading. Please be clear what you mean and observe to avoid misunderstandings.

Line 363: "the 1-m thick FYI floe" –> In lines 241-244 you already provided some information about these observations. I note that these are not coherent and not specific enough. What did you mean by "in the vicinity of the 2D site" in those lines? There you gave one date (March 5), here you give two dates. The thickness of that floe (is it representative?) is just 1 m, i.e. substantially less than the surveyed floe with 1.4-1.5 m thickness. I suggest to comment in your manuscript about these differences and slight misfit in information. I also interpret from your writing that snow depth (on that single floe) increased by 0.08 m between March 5 and March 23 and that that snow depth is considerably smaller than the average or modal snow depth of the 2D survey site. Comments?

Lines 379-382: I have a stupid question in this context: How do you compute the sea-ice thickness from estimates of sea-ice freeboard and snow depth using the classical equation (e.g. equation 6) when the sea-ice freeboard is negative? Then the first term in equation 6 is negative. Is the retrieval using that equation defined at all?

Line 383++: "Our study shows that saline snow conditions can ..." –> I suggest to formulate more clearly how the conditions met in this study differ from those encountered on Canadian Arctic fast ice. There the snow cover was dry, the sea-ice freeboard positive and the brine concentration in the basal snow layers solely caused by the high sea-ice salinity near the ice-snow interface. Here, during N-ICE2015, the situation appears to have been completely different, with a substantial amount of negative sea-ice freeboard, hence flooding of the ice-snow interface and an (unknown?) amount of slush at the ice-snow interface from which large amounts of brine can be wicked up (how high?) into the overlying snow.

Line 407: Please explain why there are two different snow thickness values (here in the text and also in Fig. 9 a)

Line 408: I suggest to add a statement about the fact that the survey cite is located in an area where the shown CS-2 products (and also the snow depth) show large spatial variability. In light of the fact that sea ice is not static but drifts, your "verdict" about

the quality of the CS-2 product could perhaps formulated in a less harsh way. I note in this context, that you completely ignored the CPOM results published by Tilling and co-workers, and issue which I kindly ask you to amend in your manuscript.

Typos / editoral remarks:

Line 24: "wicking and saturation into" –> "wicking into and saturation of". Later in the sentence: Would it be sufficient to write "causing the airborne radar signal ..."? I would read more fluently. If "more diffuse scattering and influenced" shall be kept then I suggest to split the sentence into two.

Line 33: "it may result ..." –> I am not sure I understand what this refers to.

Line 43: You could add a sentence stating the importance of the thickness of the snow layer on sea ice on the in-ice and under-ice biological processes.

Line 73: I suggest to cite the work of Willat et al. (2010) here, also from the Antarctic but a different region: Digital Object Identifier 10.1109/TGRS.2009.2028237

Line 85: Perhaps switch "Atlantic Sector of the Arctic Ocean" and "Southern Ocean" since your primary focus is in the Arctic Ocean?

Lines 112/113: "... accuracy ... higher ... " –> I know what you mean but a reader might stumble at first glance being surprized that the EM31 accuracy is "better" for rough and deformed ice - which, I guess, quite some readers automatically imply into "higher". Perhaps it might make sense to write "worse"?

Lines 122/123: "We use the results of the independent snow transects from Floe 2 to provide the regional context ..." –> I am a bit confused. You have that 400 m by 60 m survey area on the floe. And then you have those additional (>5000) measurements within 5 km of the ship. These are - at least this is my assumption - not necessarily on floe 2. But these are the measurements which provide the regional context, am I right?

Figure 2: What is "WAV snow depth"?

Line 153: "During the survey ..." –> Does this refer to the OIB survey or to the ground-based survey?

Line 161: "..., and the ..." –> delete "and"

Line 171: Any measurements of the density of second-year ice?

Line 178: "for flooding" –> "to flooding"

Line 190: "can calculated" –> "can be calculated"

Line 250: "second mode" –> "mode"

Line 254: "0.08 m than" –> "0.08 m larger than"

Lines 263-265: "with the ... of the ship" is kind of a repetition of the end of the previous paragraph. I suggest to simply refer to the above-mentioned transects.

Line 273: "(FB2)" can be deleted, I guess.

Line 287: "of+-0.06 m" –> two spaces are missing

Line 290: "Figure 5" –> "Figure 6" "lie in the negative range, to -0.1 m." –> "are negative with magnitudes up to 0.1 m."

Line 291: "Results in the same range, with ... 0.09 m, are obtained ..." –> "Results in the same range are obtained ... elevation, resulting in an average value of hfb4 ... 0.09 m (FB4, see Figure 6)."

Line 296: "... elevation value" –> a good place to refer to equation (7).

Line 305: "asfor" –> "as for"

Lines 305/306: "calculated from a combination ..." –> can you refer to one of the equations? Also, you used "SP" to denote the snow probe measurements repeatedly and can do so here as well.

Figure 7, caption: Line 326: This second "SR" needs to be "SP". I suggest to add a

note that the annotation of the color bar is non-linear.

Line 356: "pushes" –> "can push" as this is a function of ice thickness. I also strongly recommend to split this long sentence into at least two sentences.

Line 381: "of vertical shift" –> "of a vertical shift"

Line 382: "... horizon, caused ... freeboards." –> perhaps better: "... horizon, caused by slush above the ice-snow interface associated with the negative freeboard and additional wicking up of brine into the overlying snow."

Line 391: "from the sea ice surface" –> "from the sea ice or slush surface"

---

## Author Comment (AC1) · 18 Jan 2021

Dear Reviewer #1,
We are very thankful for the positive feedback and the valuable comments.
We revised our manuscript accordingly and tried to address your comments carefully. Please find our responses to your comments in blue below.

Thank you again and best regards,
Anja Rösel on behalf of the co-authors.

Major comments:
-My main concern has to do with the writing of some sections of the paper. Indeed there are several confusing long sentences throughout the manuscript that challenge the overall understanding of the paper. I have to point out paragraph 2.2 (Groundbased measurements), which is really dense and needs to be re-written. I suggest that the authors add sub-sections.
Thanks a lot, we revised the manuscript accordingly, created subsection in the 2nd paragraph and re-wrote the discussion/conclusion part. We think it will help the overall understanding of the paper.

-The notation used for the different observation techniques and instruments is confusing. This results in equations using variables with too many subscripts (equation 6 for example). I suggest that the authors use only the name of the instruments in the definition of variables. This should greatly improve the quality of the paper.
We revised the variable naming according to the table here. This was also suggested by the 2nd reviewer.

| old name | new name | alternates | what it means |
|---|---|---|---|
| ht | ht | | combined snow depth + sea ice thickness |
| ht1em | $ht_{EM}$ | | combined snow depth + sea ice thickness measured by EM |
| **Total (snow + ice) freeboard / snow freeboard** | | | |
| hfbs | hfbs | F | generally |
| hfbs2is | $hfbs_{is}$ | | ...from drill holes |
| hfbs3atm | $hfbs_{ATM}$ | | ...from laser scanner (ATM) |
| **Sea ice thickness (the total ice component)** | | | |
| hi | hi | Zi | generally |
| hi2is | $hi_{is}$ | | ...from drill holes |
| hi1is | $hi_{EM,SP}$ | | ...estimated from EM and snow probe |
| hi4atm,sp | $hi_{ATM,SP}$ | | from ATM total freeboard, snow probe depths and densities |
| **Snow depth** | | | |

| | | | |
|---|---|---|---|
| hs | hs | Zs | generally |
| hs2is | $hs_{is}$ | | ...from drill holes or snow pits |
| hs1sp | $hs_{sp}$ | | ...from snow probes |
| hs3sr | $hs_{sr}$ | | ...from radar |
| **Ice freeboard (no snow)** | | | |
| hfb | | Fi | generally |
| hfb2is | $hfb_{is}$ | | ...from drill holes |
| hB3atm,sr | $hfb_{atm,sr}$ | | ...from ATM and snow radar |
| hfb1is | $hfb_{em,sp}$ | | ...estimated from EM and snow probes |
| hfb4atm,sp | $hfb_{atm,sp}$ | | ...estimated from lidar and snow probes |

Minor comments:

P2L39-42: the sentence" On the other hand $: : :$. melting" needs to be rephrased.
Perhaps add parenthesis between "compared to" L40 and Perovic, 1996).
- Rephrased to  "On the other hand, in spring and summer, snow reflects with its high optical albedo in the range of 0.7-0.85 short-wave radiation and prevents the underlying sea ice with an albedo of about 0.6 from melting (Grenfell and Maykut, 1977, Perovich, 1996)."

P2L41: Remove capital letter on "Snow reflects".
- done

P2L56: Replace "snow/sea ice" by "snow/ice".
- done on several places throughout the document to keep it consistent

P3L75: Please specify which freeboard you are referring to (snow/ice).
- we changed it to 'ice  freeboards'

P4L101: I suggest adding a link or reference to Snow-Hydro.
- We think this is here not necessary, the company is known in the community.

P11L265: Add a comma between "site" and "the snow distributions".
- done

P13L289: Replace "0.00" by 0.
- done

P14L305: Replace "asfor" by "as for"
- done

P14L311: Avoid using "result" twice in the same sentence, I suggest to remove "the resulting".
- Done

P14L311-311: Please revise:" Our results suggest $: : :$ two-layer snow situation".
- rephrased to: 'Our results suggest that the resulting bias is approximately 0.4 m. This is consistent with the result shown in Figure 6, which indicates a bias of approximately 0.03 m in FB1. This bias

might be caused by the same reason (i.e., eqn. 2 also does not account for the two-layer snow situation).'

P16, Figure 7: Please correct the legend: FB3 and FB4 are the same.

- Typo in the caption corrected and changed according the new labeling scheme.

P17L345: replace "about the R/V" by "from the R/V".

- done

P18L352: Seems to be an extra space between "undergo" and "absorption".

- solved

P18L364: Remove "of" in "of > 11 psu"

- done

P18L371-372: Seems to be an extra space between "thereby" and "impacting".

- solved

P20:L415: replace "should to be" by "should be".

- done

---

## Author Comment (AC2) · 19 Jan 2021

Dear Reviewer #2,
We are very thankful for the positive feedback and the valuable comments.
We revised our manuscript accordingly and tried to address your comments carefully. Please find our responses to your comments in blue below.

Thank you again and best regards,
Anja Rösel on behalf of the co-authors.

GC1: The study shines through an excellent set of observations. There are parts of the description which ask for improvement, though. One is an improved consideration / discussion of snow depth and sea-ice thickness variations between survey site (local), 5 km circle around Lance (extended in-situ survey) and 10 km circle around Lance (OIB); more details regarding this issue I give in the specific comments. The discussion about scaling and representativity issues is light and could be expanded; elements of this topic I try to express with the following questions. Can one really, as done, combine the OIB data with the in-situ data with such high accuracy? Are EM31 total thickness measurements really that accurate that one can derive highly accurate sea-ice freeboard when combining this data with snow probe snow depth observations? How does the EM31 signal respond to a possibly spatially extended area of flooded sea ice?
We looked again at the data, matched the snow radar measurements with the closest SP data and re-calculated and re-made the Figures 5-8 and improved with it also the discussion about the scaling (local and regional). And yes, we think it's possible to connect OIB data (10s of meter range of the snow radar) with very high-resolution data like SP (cm-range) and a bit coarser EM data (m-range) to get knowledge of the accuracy - therefore we do in addition the inter-comparison with regional ground data (5 km scale) to get the averages of snow depth and not the single point data. Also, the comparison as a pdfs allows to compare different scaled data.

How does the issue that OIB is known to underestimate thin snow / has issues over highly deformed sea ice influence your results? Finally, the one (or maybe two) ice salinity profiles used at the end to conclude that your observations prove that saline snow can be the main cause for the observed biases, are not very well connected to
the rest of the manuscript - even though they are the dominant topic in the discussion and conclusion section. I recommend to give the description of these in situ observations more weight and demonstrate more clearly how well (hopefully) these single point measurements represent the conditions in the detailed survey area.
We rewrote and restructured the discussion and result section to get this more connected. We also moved the Figure of the salinity profile from the Appendix in the main manuscript.

GC2: My impression is that the MAIN issue is that OIB data result in a sea-ice freeboard bias of 0.2 m. This is a SUBSTANTIAL bias but it is not presented and discussed in an overly prominent way in the manuscript. I recommend to be more exhaustive in the discussion of particularly these results in the context of Figures 6 through 8.
We looked again at the data and have to reinterpret statements about a 0.2 m bias. Looking regionally we see a 0.06m difference between OIB radar and MP data (Figure 5b), although it is very difficult to connect these two datasets or identify any features which might explain the pretty good fit between them. The 2D field is the only location where we can reliably pin OIB snow radar to near-coincident in situ observations. Using only radar data inside the 2D field we see a much closer fit between in situ and radar snow depths - more realistically in line with expected brine wicking or flooding depths and also closer to regional values.

GC3: The secondary issue is the snow density of the proposed () two-layer snow setup which is discussed in the context of Figure 8, hi4. There are, to my opinion, at least two areas of improvement. The first one is superposing the drill-hole sea-ice freeboard data onto Figure 7 and discuss the results. The second one is to conduct a sensitivity study which plays around with possible snow layer depths and densities used in

Equation 6 to derive hi4. A pre-requisite for these actions is an appropriate introduction of what you cann the two-layer snow setup, which is not adequately described yet in the manuscript.

In this context, I also kindly ask for clarification of the wording "slushy basal snow layer" versus "snow-ice basal snow layer" because snow-ice is refrozen slush hence hard while slush is soft - which has implications for the snow probe measurements and the interpretation of the measurements.
We revised this in the manuscript to be clear. See also your specific comment for L. 357

Specific Comments:
Line 179++: I am not sure your choice of denoting different variables with 1, 2, 3, 4, using hi (I) and hs (S) and hfb (Fi) and hfbs (F) to name the variables, i.e. without subscripts, and introducing subscripts such as EM, SP or IS is an optimal solution. In any case I recommend to use it in a consistent way. This means that also in the running text hi and hs should be given as used in the formulas (see Lines 174/175 for instance). In addition, I am wondering whether it is necessary to abbreviate "in situ" with IS. To me this is confusing in the zoo of short names. But this is of course your choice. If you keep IS then you need to move its definition from Line 194 to Line 179.
We have removed numbers from all variables and made a naming scheme we think adds some clarity. In this new scheme, the subscript 'IS' is used to mean 'drill hole data', with all other sources using subscripts to denote the instruments used to collect data which values are derived from. We agree, there are many ways to name variables relevant to altimetry on sea ice, and welcome feedback which points out where clarity is needed.

Equation 2: I am sure that the freeboard in this equation needs to be the snow freeboard, i.e. hfbs.; this is the classical equation to derive sea-ice thickness I from total (sea ice + snow) freeboard F and snow depth S, isn't it?
The authors agree with the reviewer. We have corrected the equation to using the total (snow+ice) freeboard.

Then equation (3) would be the total freeboard as well and not the sea-ice freeboard. In order to end up with the sea-ice freeboard here I suggest to use the retrieval equation used for radar altimetry (in your notion):
hi1IS = (hs1SP * rho_s + hfb1IS * rho_water) / (rho_water - rho_ice)
Solved

In Line 194 it needs to be "1" instead of "2" in the variable name.
Solved

Equation 4: In the context of this equation I'd like to note that you are not consistent with Figure 3. There you denote snow freeboard as hfbs while in Equation 4 and in Line 202 you write hsfb. I like hsfb more.
The authors agree with the reviewer. We have reformulated the equation notations.

Line 206/207: While the accuracy for the drill-hole measurements is clear from the stated measurement accuracy of the measurement device, I am wondering whether you might want to add one sentence about the way you estimated the value of 0.06 m for the combination of snow probe and EM31 measurements.
We added here: "As described in Rösel et al., 2018 the uncertainty of in situ freeboards hfbEM,SP and hsfbEM,SP resulting from the propagation of uncertainties in the snow and ice densities and the sampling uncertainty (represented by the spatial variability) is estimated to be on average ±0.06 m."

Equation (7): I suggest to split this equation into two. Please first introduce hB without the interpretation that it can be decomposed into sea-ice freeboard, basal snow-layer thickness and an error term and subsequently, in a follow-on equation (if necessary) show the decomposition of hB. I note, after having read

the entire manuscript, that there is limited reference to and usage of this equation. Or did I overlook a figure or table where you provide an estimate of the basal snow-layer thickness?
We revised the manuscript and removed Eq (7).

Figure 4: I am a bit confused with what I see here. I appears to me that this plot starts on the left somewhere close to the top right corner of the map shown in Figure 2, takingone of the overpasses shown towards the southwest (crossing the survey area). Fine. Then comes the aircraft turn; the main part of the radar-echogram shown in Figure 4 is related to sea-ice conditions "on ice floe south of the survey site". Now, what I have problems with is the fact that you stated the times for overpass #2 and #3 as 15:37 and 15:43 in the caption of Figure 4 while in the heading line just above the figure it specifies a time range around 15:43, suggesting that this is only overpass #3. Could it hence be that we only look at overpass #3 and that only the small part (left quarter of the echogram) is coincident with the tracks (actually only overpass #3) denoted in Figure 2? If this is the case, I am wondering whether it wouldn't make sense to expand that left quarter of the echogram because this has a direct relation to the survey area.
This is solved; it is only overpass #3 and the survey area is now indicated by a red box

Lines 253-255: "In addition ..." How important is it to explicitly mention the mean of these very few (compared to the other samples) snow depth observations at the drill hole sites in the text? Would it be sufficient to only show this value in the Table? I am suggesting so because you are coming up with quite a number of different snow depth values and it begins to become confusing. Particularly, since you repeat this information in Line 273.
The authors agree with the reviewer. We have removed the sentence and directed the readers to Table 2 in the appendix and Figure 5.

Line 263: What is the motivation to draw a 10 km circle around R/V Lance for the air-borne data when the in-situ observations along the five transects were carried out within a 5 km radius circle?
We chose the 10 km radius for the snow radar data to show the regionality, the in situ transects were only in a radius of 5 km around the ship. Internal data comparison of the snow radar data data do not show significant differences if we choose a 5 km or 10 km radius.

Line 275/276: "Three ... before drilling." –> How do you know?
To avoid confusion, we have rephrased the sentence to 'Three out of the ten drill-holes were found to be flooded.

Figure 5: In the left panel you denote hs1 and hs3 with the respective methods while in the right panel you write "regional". My suggestion would be to be consistent. At this point the reader possibly knows that hs1 is based on snow probe data and that hs3 is based on the snow radar. Hence you could use "survey area" in the left panel and
keep "regional" in the right panel.
Thanks for the suggestion, we revised the figure and changed the labeling accordingly.

Lines 294-298: These sentences about brine wicking etc. fall a bit from heaven. I suggest to start a new paragraph here and motivate these further considerations by again stressing that near-zero ice-freeboard supports flooding of the ice-snow interface and subsequent upward wicking of seawater and/or brine into the basal snow cover. The paragraph about the c-profile of the salinity given at the beginning of Section 3 could be placed here in a much more logical way - as this piece of information seems a bit lost where it is located currently.
Thanks for the comment. We have reorganized the introduction of snow salinity caused from flooding and brine wicking in a separate paragraph. We have also moved the salinity c-profile to the discussion section.

Figure 7: I suggest to overplot the freeboard values shown in Table 2 of the appendix onto the maps in each panel by using, e.g., a color-filled circle, and discuss what you see.
Figure 7 was re-done and might be clearer now.

Lines 303-313 / Figure 8:
- Line 305: I don't find hi_REGIONAL in Figure 8.
This now changed to hiATM,SR(all) (also in the updated Figure)

- Line 307: "slightly thinner" –> It might be a matter of taste but a difference of 0.3 to 0.4 m in a thickness range between 1.1 and 1.5 m I would not call "slightly".
We have removed 'slightly' in the revised manuscript.

In addition, those 5 regional survey lines, were these laid out without checking the ice conditions beforehand?
No, this was done afterwards. They were a combination of first year ice and second year ice. We tried to stick to a triangular shape, to cover randomly everything without pre-choosing the ice type.
What I mean by this is, that based on the large sea-ice thickness standard deviation it seems likely that these 5 lines had a substantial fraction of thin ice from refrozen leads while the bulk of the sea ice inbetween might have had a similar thickness than the ice of the survey site. Please check and, if need be, re-phrase statements.

- I am missing commenting on hi3.
Thanks for the comment. We considered this.

- Line 309-313: I suggest to re-write this part. hi4, if computed using Eq. 6, uses the sea-ice freeboard (hfb) which is derived from airborne ATM total freeboard (accurate) and the in-situ snow depth (accurate as well); in addition, for the snow part it uses (again) the in-situ snow depth. The densities you used are those which you measured in the field = accurate. Hence, at first glance it is not clear why even with the densities you measured the hi4 is biased compared to hi1 and hi2. The statement that the bias in sea-ice thickness between hi1 and hi4 of about 0.4 m is consistent with the bias in hfb of 0.03 m and that this is in part due to the not sufficiently well considered "two-layer snow set up" is not backed-up yet by your writing or the figures. If, as you suggest, snow densities across this two-layer snow set up are the main cause, then I strongly suggest to play around with potential snow densities (you measured some in the field, didn't you?) and layer thicknesses to figure out whether your hypothesis is true. In other words: Which snow density values of the proposed two-layer snow setup explain
the observed difference between hi1 and hi4? Are these realistic?
Good point. We recomputed ice thicknesses for figure 8 and found a much closer match between datasets.

Discussion section:
General comments: I don't see a reason why to write "total snow depth" or "total snow thickness". I guess "total" can be omitted.
Thanks. "Total" omitted before snow. Also, throughout the manuscript, we have replaced snow thickness by snow depth.

Line 345: This is a reminder that it might make sense to provide a scientific motivation for using a 10 km radius for the airborne data versus a 5 km radius for the in situ data. If by chance the airborne data observed a comparably large fraction of thin ice, then the airborne snow depth would naturally be smaller and hence its bias to the in situ snow depth.

As mentioned in our reply to a comment above we did some analysis of a 10km vs a 5 km radius of the airborne data which do not show significant differences.

Line 357: "slushy, snow-ice formation in the basal layers of the snow pack" vs. Line 367: "formation of highly saline and saturated slush in the basal snow layers" –> It is a difference whether one speaks of wet / saturated slushy snow ... which is a rather soft material, or whether one speaks of snow-ice which - at least to my understanding - is refrozen ... and hence hard. It is, to my opinion, important to distinguish between those because I'd think that the SP measurements would penetrate slush and hence INCLUDE the thickness of the slush layer into the snow depth reading while these would not penetrate snow ice and exclude that part from the snow depth reading. Please be clear what you mean and observe to avoid misunderstandings. As already mentioned above, we clarified and revised this already.

Line 363: "the 1-m thick FYI floe" –> In lines 241-244 you already provided some information about these observations. I note that these are not coherent and not specific enough. What did you mean by "in the vicinity of the 2D site" in those lines? There you gave one date (March 5), here you give two dates. The thickness of that floe (is it representative?) is just 1 m, i.e. substantially less than the surveyed floe with 1.4-1.5 m thickness. I suggest to comment in your manuscript about these differences and slight misfit in information. I also interpret from your writing that snow depth (on that single floe) increased by 0.08 m between March 5 and March 23 and that that snow depth is considerably smaller than the average or modal snow depth of the 2D survey site. Comments?
Thanks for the comment. The ice salinity measurements are from March 5. We have removed March 23 in the sentence, in the revised manuscript.

Lines 379-382: I have a stupid question in this context: How do you compute the seaice thickness from estimates of sea-ice freeboard and snow depth using the classical equation (e.g. equation 6) when the sea-ice freeboard is negative? Then the first term in equation 6 is negative. Is the retrieval using that equation defined at all?
No, the second term would always be positive (and with a higher number since it uses snow depth). For the ice thickness calculations, we used eqn 2. Eqn 6 was not used and removed from the manuscript to avoid confusion.

Line 383++: "Our study shows that saline snow conditions can ..." –> I suggest to formulate more clearly how the conditions met in this study differ from those encountered on Canadian Arctic fast ice. There the snow cover was dry, the sea-ice freeboard positive and the brine concentration in the basal snow layers solely caused by the high
sea-ice salinity near the ice-snow interface. Here, during N-ICE2015, the situation appears to have been completely different, with a substantial amount of negative sea-ice freeboard, hence flooding of the ice-snow interface and an (unknown?) amount of slush at the ice-snow interface from which large amounts of brine can be wicked up (how high?) into the overlying snow.
Thanks for the comment. We have rephrased the previous section (before Line 383) differentiating between landfast FYI (with positive freeboard) and ice with negative freeboards. There were no direct measurements of how much slush there was nor the slush salinity. Nandan et al. (2020) (Figure 4b) reported high salinities towards the basal snow layers (up to 25 ppt), sampled during the N-ICE 2015 campaign. However, we cannot confirm if these high basal salinities are caused by brine wicking from the underlying slush layers.

Line 407: Please explain why there are two different snow thickness values (here in the text and also in Fig. 9 a)
The two different values result from the Warren climatology used for the Kurtz algorithm. We rephrased as follows: 'Modelled snow depths of 0.15 m and 0.37 m (derived from Warren et al., 1999; Kurtz et al., 2014, Figure 9a) are underestimated when compared to the observed in situ snow depth, which averaged 0.55 m.

Line 408: I suggest to add a statement about the fact that the survey cite is located in an area where the shown CS-2 products (and also the snow depth) show large spatial variability. In light of the fact that sea ice is not static but drifts, your "verdict" about the quality of the CS-2 product could perhaps formulated in a less harsh way. I note in this context, that you completely ignored the CPOM results published by Tilling and co-workers, and issue which I kindly ask you to amend in your manuscript.

Thank you for the comment. We have rephrased the paragraph with more constructive criticism for the CS-2 product quality and have also referred to Tilling et al. (2018) where their work showcases the impact of sea ice drift affecting the quality of CS-2 products.

"However, presently operational CryoSat-2 retracker algorithms or empirical models (e.g. Hendricks et al., 2010; Ricker et al., 2014; Kurtz et al., 2014) do not account for snow pack flooding as a source of error, affecting the accuracy of sea ice freeboard and thickness estimates. Moreover, since our survey site was also drifting, we acknowledge the impact of sea ice dynamics also affecting the correlations between in situ measurements and satellite-derived estimates, both acquired at different times (Tilling et al., 2018)."

**Typos / editoral remarks:**

Line 24: "wicking and saturation into" –> "wicking into and saturation of". Later in the sentence: Would it be sufficient to write "causing the airborne radar signal ..."? I would read more fluently. If "more diffuse scattering and influenced" shall be kept then I suggest to split the sentence into two.

Thanks for the comment. The sentence is split into two as follows: These conditions caused brine wicking into and saturation of the basal snow layers. This causes the airborne radar signal to undergo more diffuse scattering, which results in the location of the radar main scattering horizon to be detected well above the snow/sea ice interface. This leads to a subsequent underestimation of total snow depth, if only radar-based information is used.

Line 33: "it may result ..." –> I am not sure I understand what this refers to.

We have revised the sentence as follows: "Our results suggest that sea water flooding of the snow/ice interface leads to underestimations in snow depth or overestimations of sea ice freeboard, measured from radar altimetry, in turn impacting the accuracy of sea ice thickness."

Line 43: You could add a sentence stating the importance of the thickness of the snow layer on sea ice on the in-ice and under-ice biological processes.

Thanks for the comment. We have added a sentence in the revised manuscript as follows: "In addition, snow cover controls the amount of transmittance of photosynthetically active radiation affecting the productivity of primary algae and phytoplanktons (Mundy et al., 2007)."

Line 73: I suggest to cite the work of Willat et al. (2010) here, also from the Antarctic but a different region: Digital Object Identifier 10.1109/TGRS.2009.2028237

Willatt et al., 2009 added to the citation.

Willatt, R. C., Giles, K. A., Laxon, S. W., Stone-Drake, L., & Worby, A. P. (2009). Field investigations of Ku-band radar penetration into snow cover on Antarctic sea ice. IEEE Transactions on Geoscience and remote sensing, 48(1), 365-372.

Line 85: Perhaps switch "Atlantic Sector of the Arctic Ocean" and "Southern Ocean" since your primary focus is in the Arctic Ocean?

Changed.

Lines 112/113: "... accuracy ... higher ... " –> I know what you mean but a reader might stumble at first glance being surprized that the EM31 accuracy is "better" for rough and deformed ice - which, I guess, quite some readers automatically imply into "higher". Perhaps it might make sense to write "worse"?

We have deleted the accuracy phrase for rough and deformed ice to avoid confusion.

Lines 122/123: "We use the results of the independent snow transects from Floe 2 to provide the regional context ..." –> I am a bit confused. You have that 400 m by 60 m survey area on the floe. And then you have those additional (>5000) measurements within 5 km of the ship. These are - at least this is my assumption - not necessarily on floe 2. But these are the measurements which provide the regional context, am I right?
Thanks for the comment. We have rephrased the sentence in the revised manuscript, as follows: "We use the 2-D grid snow depth measurements and those sampled via transects within a 5 km radius, to provide a spatial representativeness and context from local- to regional-scales".

Figure 2: What is "WAV snow depth"?
WAV refers to the snow radar-derived snow depths using the NOAA Wavelet technique (Newman et al., 2014). We have added a sentence in the figure caption on this.

Line 153: "During the survey ..." –> Does this refer to the OIB survey or to the groundbased survey?
We changed it to 'During the OIB survey'

Line 161: "..., and the ..." –> delete "and"
done

Line 171: Any measurements of the density of second-year ice?
We do not have any confirmed and accurate sea ice density measurements for second year ice from N-ICE.

Line 178: "for flooding" –> "to flooding"
done

Line 190: "can calculated" –> "can be calculated"
done

Line 250: "second mode" –> "mode"
done

Line 254: "0.08 m than" –> "0.08 m larger than"
Done

Lines 263-265: "with the ... of the ship" is kind of a repetition of the end of the previous paragraph. I suggest to simply refer to the above-mentioned transects.
We have deleted the repetition of the shup usage and referred them to the transect measurements.

Line 273: "(FB2)" can be deleted, I guess.
Deleted.

Line 287: "of+-0.06 m" –> two spaces are missing
Space added.

Line 290: "Figure 5" –> "Figure 6" "lie in the negative range, to -0.1 m." –> "are negative with magnitudes up to 0.1 m."
Rephrased.

Line 291: "Results in the same range, with ... 0.09 m, are obtained ..." –> "Results in the same range are obtained ... elevation, resulting in an average value of hfb4 ... 0.09 m (FB4, see Figure 6)."
We have rephrased the sentence in the revised manuscript as suggested.

---

## Author Response (AR2)

Dear Editor,

Thanks a lot for giving us the opportunity to revise the manuscript for another round.

Please find below our responses for Referee #2. We are thankful for the positive feedback and the valuable comments.

We revised our manuscript accordingly and tried to address the comments of the referee carefully.

Please find our responses to his comments in blue below.

Thank you again and best regards,

Anja Rösel on behalf of the co-authors.

**Response for Referee #2:**

Dear Stefan,

Thanks a lot for your very valuable comments and thoughts throughout the review process. Your input is as always highly appreciated.

Please find below our responses to your comments in blue.

Thanks again and best regards,

Anja and all co-authors

*Review of the revised version of*

*Implications of surface flooding on airborne estimates of snow depth on sea ice*

*by Rösel, A., et al.*

*I really like this paper and appreciate that the authors invested the effort to revise their manuscript successfully.*

*The editor was so kind to invite me to have a second look, which I was delighted to do. Please find below my comments which you might consider to take into account before finally submitting this interesting contribution.*

*The majority of my comments point to minor edits or technical issues. I warmly recommend to once again cross-check usage of acronyms and subscripts, for instance.*

*There are two things I have more problems with.*

*The first one might be based on myself having misunderstood parts of the theory applied here. It deals with Equations 2 and 3, i.e. how you went from one to the next. I detailed this in my comment further below. Either there was just a typo. Or I misunderstood something. Or you forgot to take things into account adequately (in which case I don't know whether you might need to carry out some re-calculations.*

Very good point and thanks for spotting it. It is indeed a typo in the equation and in line 215 (hfb2_IS instead of hfbs_EM,SP), resulting from the replacement of the variable renaming. This is corrected now.

*The second one is perhaps a matter of taste. It deals with (my observation) the issue that Figures 5, 6, and 8 show so much more than you are actually commenting and discussing about and I am wondering whether you left out a few (those, see my comments) issues on purpose because you wanted to keep the paper short. In other words, I believe that the discussion section, which focuses a lot on the impact of a basal saline (and slushy) snow layer on radar observations - be it the OIB snow radar or be it an altimeter, could be tied much more to the very obvious discrepancies between in-situ snow depth observations and radar-derived snow depths - including the impact this has on freeboard estimation and freeboard-to-thickness conversion.*

*Finally, because of the substantial impact a basal saline (and slushy) snow layer apparently exerts on the radar observations, I felt that the notion given in the abstract about the overall differences between OIB snow depth and in-situ observations at both spatial scales considered (0.12 m and 0.06 m) could easily be misinterpreted. You stated on your own somewhere in the paper that the regional difference of 0.06 m, based on a comparison of much more diverse in-situ observations taken from 5 days within a several weeks timeframe with OIB measurements obtained within minutes on one particular day, is possibly not as trustworthy as the results obtained for the 2D field survey. In addition, the effect of the obvious snow depth under-estimation by the OIB radar becomes particulary striking in Fig. 6 and 8. Therefore, wouldn't it be a reasonable idea to stress that the observed (comparably small) snow depth differences and their downstream effects are a result of the combination of one part of the data having excellent agreement and one part of the data where the differences are substantial? This paper has the potential to clearly communicate this issue to the community. While this might be a minor issue for the Arctic, it has widespread consequences for Antarctic sea-ice thickness retrieval attempting to use satellite radar altimetry.*

Thanks for the two comments above. We included the topic of the different scales and the different temporal resolution again in the discussion section (L.424-427 and 463-466)

*Line 74: I am a bit confused with the year used for the citation Willatt et al.; I agree that the publication date is October 9, 2009, but the paper appeared in the first issue of the TGRS volume of the year 2010: 48(1). You might want to check which is more appropriate. Personally, I would go for the printed version in this case but perhaps policies have changed?*

You are right: official citation is Willatt, 2010; we updated it accordingly.

############

*Table 1: Would it make sense to replace "combined snow depth + sea ice thickness" by "total (snow + ice) thickness" as is also used for the freeboard?*
yes, updated

*I suggest to replace the "/" by an "or" to avoid confusion with the mathematical symbol. Alternatively you could also write "(also: snow freeboard)" instead of "/ snow freeboard"*
Yes, thanks. We go for the 2nd option.

*You could add "(IS for in situ)" behind "from drill holes"*
yes, updated

*The "(the total component)" seems not to be required? Please check whether this supplementary information adds clarity.*
It's deleted. It does not add further information.

*"from radar" could be expanded into "from snow radar" so that the subscript "SR" matches better.*
yes, updated

*The "(no snow)" could also be considered superfluous because you described the term "total freeboard" above already. But this is clearly up to you.*
It's deleted. It was more a note to ourselves which categories we have

*In the last row you might want to replace "lidar" by "ATM" for consistency.*
Yes, right.

############

*Line 190: I suggest to add that you used the same sea-ice density for SYI as you used for FYI (it that applies).*
I changed FYI to sea ice.

*Should $hi\_IS$ in Eq. (1) be replaced by $hi\_{EM,SP}$? [note that I use "_" to denote a subscript]. Otherwise I don't understand the sentence before. $hi\_IS$ denote the sea ice thickness from in situ measurements, doesn't it?*
Sure, thanks for noticing it.

*Equations (2) and (3): I have a problem to understand how Eq. (2), which contains sea ice thickness (from EM+SP i.e. possibly the one just computed in Eq. 1), total freeboard (to be derived) and snow depth (from SP), you compute the ice freeboard $hfb\_{EM,SP}$. To me Eq. (3) would make sense if you'd write $hfbs\_{EM,SP}$*
Thanks – it was 'just' a typo as stated above

*If you insist that Eq. (3) refers to ice freeboard ... then something went wrong from Eq. (2) to Eq. (3) because:*

*$hi\_{EM,SP} = hfbs\_{EM,SP} * rho\_w/(rho\_w-rho\_i) - hs\_SP * (rho\_w-rho\_s/rho\_w-rho\_i)$ -->*

*$hi\_{EM,SP} * (rho\_w-rho\_i)/rho\_w = hfbs\_{EM,SP} - hs\_SP * (rho\_w-rho\_s)/rho\_w$ -->*

*$hi\_{EM,SP} * (rho\_w-rho\_i)/rho\_w = hfb\_{EM,SP} + hs\_SP - hs\_SP * (rho\_w-rho\_s)/rho\_w$ -->*

*$hi\_{EM,SP} * (rho\_w-rho\_i)/rho\_w = hfb\_{EM,SP} + hs\_SP * (1 - (rho\_w-rho\_s)/rho\_w)$ -->*

*$hi\_{EM,SP} * (rho\_w-rho\_i)/rho\_w = hfb\_{EM,SP} + hs\_SP * rho\_s/rho\_w$ -->*

*$hi\_{EM,SP} * (rho\_w-rho\_i)/rho\_w - hs\_SP * rho\_s/rho\_w = hfb\_{EM,SP}$ -->*

*$( hi\_{EM,SP} * (rho\_w-rho\_i) - hs\_SP * rho\_s ) / rho\_w = hfb\_{EM,SP}$*

*... which is not your Eq. (3). I am happy to learn, however, that the step hfbs_EM,SP = hfb_EM,SP + hs_SP is not correct.*

*Lines 220/221: I guess the years of the references need to be in parentheses here.*
Corrected

*Line 227-229: Should, in line 227, hfb_EM,SP be hfbs_EM,SP? I am confused to see the subscript "IS" in line 229, where you in fact write about drill hole measurements which, according to your table 1 get a subscript "IS" (= correct), while in line 227 where you write "in situ freeboards" you appear to mix the general notation (= no subscript) with the one for the combined EM / snow probe data (_EM,SP). I am still not there, sorry.*
The two sentences in line 227-229 are about uncertainties of the different freeboards; We revised it to make it understandable: "As described in Rösel et al., 2018 the uncertainty of the ice freeboard $hfb_{EM,SP}$ and the total freeboard $hfbs,$ resulting from the propagation of uncertainties in the snow and ice densities and the sampling uncertainty, is estimated to be on average ±0.06 m. The accuracy of freeboards $hfb_{IS}$ and $hfbs_{IS}$ from the in situ drill-hole measurements is ±0.01 m (Rösel et al., 2018)."

*Line 244: It might make sense to motivate here why you compute hi_ATM,SP using the Eqs. 5 and 6 and not simply using the "classical" Eq. 2, aka: hi_ATM,SP = hfbs_ATM * rho_w/(rho_w-rho_i) - hs_SP * (rho_w-rho_s/rho_w-rho_i)*
*I guess the reason is kind clear ... because you want to stress the bias one gets using a radar altimeter in case of (refrozen) slush ... but it is not well motivated here.*
Yes, we included it in L .245 "Ice freeboard ($hfb_{ATM,SP}$) and sea ice thickness ($hi_{ATM,SP}$), including a potentially refrozen slush layer, can be derived from a combination of the airborne data measurements acquired over the 2-D survey field site with the in-situ snow-probe data…"

*Line 274: "of 0.55 m" could be deleted as both values, mean and modal, are given in Line 267.*
ok

*Line 277: "larger" --> The mean snow radar is 0.42 m and hence 0.08 m SMALLER than the snow depth at the drill-hole location of 0.50 m.*
right, corrected

*Lines 282-284: These 5 surveys were carried out during different days, nevertheless you average the snow depths and sea ice thickness values to one common value. This kind of precludes that there were not major snowfall events inbetween. Is this correct? Would it make sense to mention this in your sentence?*
This is unfortunately not correct – in deed there were snowfalls in between. Adding the early transect from February and Early March to the average snow thickness might reduce the overall average a bit, but considering the amount of precipitation (relatively small), the drift and relocation we could not see a general increase in snow depth during the sampling period, therefore we neglect the snowfall in this period. This is also stated in Rösel et al (2018): "On Floe 2, the conditions were quite stable; the average thickness of ice and snow did not change within 3 weeks of the drift on this floe." We added a sentence in the manuscript to clarify this.

*Line 287: This "snow radar" is the one on OIB, right? And the data are from the same measurement flight, i.e. March 19?*

Yes, correct; we now write: "we compared observations from the OIB snow radar measurements from the same flight within a 10 km radius around the position of *R/V Lance…*"

*Line 294: Looking back to what you just wrote: Does this mean that you consider the comparison between the OIB snow radar and the in-situ observations on the survey field more trustworthy? Would it make sense to state this?*
I would prefer to leave it open, in a sense that the reader can make up his mind him/herself: I mean it is obvious: the comparison over the survey field was done on the same day, the comparison over the larger area was done over a longer time scale. And although I state earlier that the general snow/ice conditions did not change, we could observe local changes and different behavior of snow and ice thickness; i.e. from the buoy data. And of course, we will not know if 5 km away, one day before the overflight, there was a ridge build up and is messing up our averages. This is why we mentation here the possibility of such or other events.

Figure 5: b) Isn't it interesting to see - despite the differences in the spatio-temporal sampling - that the hs_SP distribution is much narrower than the hs_SR distribution and that the latter shows a substantially smaller count of small snow depth values (< 0.3 m) than hs_SP, while the tails towards thicker snow depth values (> 0.7 m) are very similar? Will you comment on that in your discussion? --> No ... having read the discussion you do leave this uncommented. Wouldn't it be straightfoward to refer back from and forth to the discussion section when you are dealing with the upward migration of the main scattering horizon in case of a saline / slushy basal snow cover to discuss / explain the above-mentioned differences in the snow depth distributions?
In the same context I note for panel a) that the count of hs_SR values is larger than that of hs_SP for low snow depths while it is the other way round for large snow depths. Comments?

Line 305: "hfb_EM,MP"?
Sorry this is a typo/remaining from ancient time when we still named the snow probe 'Magnaprobe' – I changed it to hfb_EM,SP.

Line 306: "eqn. 3" --> Eq. 3. Please also see my comments with respect to that equation.

Table 2: I note that the value in the toprightmost cell of the table does not correspond to the respective values given in the text in Line 305 (if I am not mistaken and these two values should be the same).
The row denoted "OIB snow radar inside 2D field" should, according to what is written in Line 273, have the values 0.42 in its first cell.
I have difficulties to interpret what the row just below that is representing. Is this the regional OIB estimate? In that case the number might need to be 0.49 m (see Line 290)? I am confused.

Line 322: I see this "hi_ATM,SR(all)" for the first time here? Perhaps either introduce the "all" earlier or don't use it? I note that in Fig. 8, caption, you assign this quantity with "Pass 2"; I am confused.

Line 325-326: You could potentially explain this thinner regional mean sea-ice thickness and larger variability with the fact that the longer surveys across Floe 2 included a considerably larger thin ice fraction than the 2D survey field, am I correct?

Line 326: "average sea ice thickness" --> add "of the 2D survey cite"
If you would have computed hi_ATM,SP using the classical freeboard-to-thickness equation which employs the total freeboard (instead of the artificially introduced sea ice freeboard), what would have been the result? Since the snow probe doesn't "feel" whether the snow at the bottom is slushy or not

you should obtain fairly accurate sea-ice thickness values, am I correct?
I have difficultis to find this value of 1.90 m for hi_ATM,SP in Fig. 8. If I am not mistaken then it should be the green dashed line which denotes hi_ATM,SP (according to the legend of Fig. 8) and associated with that line I find a value of 1.52 m. What is correct?

Line 329: "a bias of approximately 0.03 m in hfb_EM,SP" --> I am confused. When I look at Fig. 6, then I see a very convincing agreement betwen hfb_EM,SP and hfb_ ATM,SP ... where is the bias? Both values match the in-situ ice freeboard measurements by 0.01 m. Perfect. The only really striking thing I see in Fig. 6 is the bimodal distribution of hfb_ATM,SR. To my opinion both, the 1st mode at -0.05 m and the 2nd mode at 0.25 m require more dedication already in the description of this figure. It might be good to find an argument / explanation for the negative 1st mode and it might be good to state that the 2nd mode is potentially indeed caused by wet / saline snow pushing the main reflecting horizon for the snow radar upwards - and refering to the discussion section where you discuss this issue. I suggest to, also in the discussion section at the respective location, refer back to Fig. 6 (and also Fig. 8) so that the reader can again check how devastating the impact of a saline / slushy basal snow layer can be in detail.

Line 329/330: While this statement is possibly correct it is perhaps more complicated than this. If we take Fig. 3, 2nd panel from left and 4th panel from left as models, then the 2nd panel would provide an in-accurate sea-ice thickness values solely because of the densities not matching (i.e. the slushly snow at the bottom having a considerably larger density than the dry snow above). While hfb is zero, hs_SP equals hfbs. Fine. In that case usage of Eq. 2 would still be ok. The 4th panel is more tricky and the inaccuracy of the sea ice thickness obtained as several reasons. The actual hfb is < 0 but the measured (by radar) hfb is zero, the measured hfbs does not equal hs_SP because the snow probe just penetrates down to the actual ice surface and hence the snow probe measures a value for hs which equals hfbs + hsil. In addition to that: the snow density for the part above the water line is fine but the snow density for the slush part is considerably larger, possibly close to the one of sea ice if not even higher than that and for sure higher than for situation described above for the 2nd panel of Fig. 3. Usage of Eq. 2 would be ... dubious.

*Figure 7: What is "MP" denoting?*
Sorry this is a typo/remaining from ancient time when we still named the snow probe 'Magnaprobe' – I changed it to SP.

*Is there any reason why you color potentially flooded areas (negative ice freeboard) in red while elevated positive freeboards appears in blue? Given the fact that flooding is associated with water I would find it more straightforward to use a reversed coloring in panels a) and b).*
I know. We had the same discussion and I guess it's a matter of taste.

*Figure 8: It is amazing how far off hi_ATM,SR is for the 2D field survey site. While hi_ATM,SR(all) at least shows a hint of the thicknes distribution you also obtained for hi_ATM,SP there is no distribution at all for hi_ATM,SR(all) ... Comments? I could not find a note in your discussion section which would get back to this issue.*
Thanks, I added a sentence into the discussion part ("However, ambiguous radar signal penetration through slushy layers (caused by sea ice flooding) and saline snow covers (caused by brine wicking from sea ice surface) may introduce a potential bias in accurate estimates of snow depth, and subsequently the resulting calculations on sea ice thickness as shown in Figure 8. In our field experiment we can clearly see an overestimation of the sea thickness, calculated from ATM surface elevation and snow radar data.")

*Line 371: I would say this is a good location to again refer to the work of Rosie Willatt et al. in both the Arctic and Antarctic (i.e. the paper you cited already plus one in the Annals of Glaciology 44 from 2011).*
Thanks, added

*Line 374: "deep snow pushes ... induces flooding" --> I would feel more comfortable with a formulation like "could" or "might induce flooding" because deep snow on thick ice is fine, for instance, and because impermeable cold ice or ice where sea water cannot enter laterally remain dry despite a negative ice freeboard.*
Right, changed.

*Line 377: "underestimate" --> "underestimation"*
Changed

*Line 378-383: These lines are written in a way that leads me to believe this is the typical situation. Is this the case?*
Yes, from my point of of view this is clearly stated by the word 'typical C-shape profile'.

*Line 383: " ." can be deleted.*
done

*Line 388: Wouldn't this be a good location to state that, strictly speaking, only remote sensing measurements involving radar are influenced by this? Neither the ATM nor the EM notice whether there is slush at the basal snow layer or whether the salinity profile in the snow has a C-shape. The same would apply to ICESat-2 by the way. Also the snow probe measurements are not influenced. Hence, only results that involve usage of hs_SR should be influenced. Is this the case?*
Yes, thanks: "…using remote sensing techniques, which involve snow radar measurements."

*Line 409: It is the other way, round, isn't it? The 0.12 m under-estimation were for the 2-D field site.*
Thanks, corrected

*Line 425/426: Please make sure the reader gets that this is a monthly average product.*
Ok, extended it to "the monthly averaged CryoSat-2 sea ice products"

*Line 427: A blank is missing before "Noticeably"*
Done